# Object-Aware Regularization for Addressing Causal Confusion in Imitation Learning

**Jongjin Park**[1][*]    **Younggyo Seo**[1][*][†]    **Chang Liu**[2]    **Li Zhao**[2]
**Tao Qin**[2]    **Jinwoo Shin**[1]    **Tie-Yan Liu**[2]

[1]Korea Advanced Institute of Science and Technology
[2]Microsoft Research Asia

## Abstract

Behavioral cloning has proven to be effective for learning sequential decision-making policies from expert demonstrations. However, behavioral cloning often suffers from the causal confusion problem where a policy relies on the noticeable *effect* of expert actions due to the strong correlation but not the *cause* we desire. This paper presents Object-aware REgularizatiOn (OREO), a simple technique that regularizes an imitation policy in an object-aware manner. Our main idea is to encourage a policy to uniformly attend to all semantic objects, in order to prevent the policy from exploiting nuisance variables strongly correlated with expert actions. To this end, we introduce a two-stage approach: (a) we extract semantic objects from images by utilizing discrete codes from a vector-quantized variational autoencoder, and (b) we randomly drop the units that share the same discrete code together, i.e., masking out semantic objects. Our experiments demonstrate that OREO significantly improves the performance of behavioral cloning, outperforming various other regularization and causality-based methods on a variety of Atari environments and a self-driving CARLA environment. We also show that our method even outperforms inverse reinforcement learning methods trained with a considerable amount of environment interaction.

## 1 Introduction

Imitation learning (IL) holds the promise of learning skills or behaviors directly from expert demonstrations, effectively reducing the need for costly and dangerous environment interaction [21, 45]. Its simplest and effective form is behavioral cloning (BC), which learns a policy by solving a supervised learning problem over state-action pairs from expert demonstrations. While being simple, BC has been successful in a wide range of tasks [4, 7, 31, 33] with careful designs. However, it has been recently evidenced that BC often suffers from the causal confusion problem, where the policy relies on nuisance variables strongly correlated with expert actions, instead of the true *causes* [11, 12, 54].

For example, when we train a BC policy on the Atari Pong environment (see Figure 1a), we observe that a policy relies on nuisance variables in images (i.e., scores) for predicting expert actions, instead of learning the underlying fundamental rule of the environment that experts would have used for making decisions. In particular, Table 1c shows that the policy trained using images with scores struggles to generalize to images with scores masked out (see Figure 1b). However, the policy trained with masked images could generalize to original images with scores, which shows that it successfully learned the rule of the environment. This implies that learning the policy that can identify the true cause of expert actions is important for stable performance at deployment time, where nuisance correlates usually do not hold as in expert demonstrations.

---

[*]Equal contribution, in alphabetical order. {jongjin.park, younggyo.seo}@kaist.ac.kr
[†]This work was done while the author was an intern at Microsoft Research Asia

35th Conference on Neural Information Processing Systems (NeurIPS 2021).

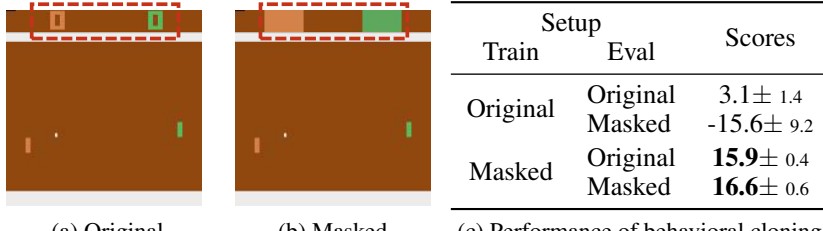

| Setup | | Scores |
| Train | Eval | |
|---|---|---|
| Original | Original | $3.1\pm{}_{1.4}$ |
| | Masked | $-15.6\pm{}_{9.2}$ |
| Masked | Original | $\mathbf{15.9}\pm{}_{0.4}$ |
| | Masked | $\mathbf{16.6}\pm{}_{0.6}$ |

(a) Original  (b) Masked  (c) Performance of behavioral cloning

Figure 1: Atari Pong environment with (a) original images and (b) images where scores are masked out. (c) Performance of behavioral cloning (BC) policy trained in Original and Masked environments, averaged over four runs. We observe that the policy trained with original images suffers in both environments, which shows that the policy exploits score information for predicting expert actions, instead of learning the underlying fundamental rule of the environment.

In order to address this causal confusion problem, one can consider causal discovery approaches to deduce the cause-effect relationships from observational data [26, 48]. However, it is difficult to apply these approaches to domains with high-dimensional inputs, as (i) causal discovery from observational data is impossible in general without certain conditions[3] [38], and (ii) these domains usually do not satisfy the assumption that inputs are structured into random variables connected by a causal graph, e.g., objects in images [29, 46]. To address these limitations, de Haan et al. [12] recently proposed a method that learns a policy on top of disentangled representations from a $\beta$-VAE encoder [19] with random masking, and infers an optimal causal mask during the environment interaction by querying interactive experts [43] or environment returns. However, given that environment interaction could be dangerous and incur additional costs, we argue that it is important to develop a method for learning the policy robust to causal confusion problem without such a costly environment interaction.

In this paper, we present OREO: **O**bject-aware **RE**gularizati**O**n, a new regularization technique that addresses the causal confusion problem in imitation learning without environment interaction. The key idea of our method is to regularize a policy to attend uniformly to all semantic objects in images, in order to prevent the policy from exploiting nuisance correlates for predicting expert actions. To this end, we propose to extract semantic objects from raw images by utilizing vector-quantized variational autoencoder (VQ-VAE) [35]. In our experiments, we discover that the units of a feature map corresponding to the objects with similar semantics, e.g., backgrounds, scores, and characters, are mapped into the same or similar discrete codes (see Figure 3). Based upon this observation, we propose to regularize the policy by randomly dropping units that share the same discrete code together throughout training. Namely, our method randomly masks out semantically similar objects, which allows object-aware regularization of the policy.

We highlight the main contributions of this paper below:

- We present OREO, a simple and effective regularization method for addressing the causal confusion problem, and support the effectiveness of OREO with extensive experiments.
- We show that OREO significantly improves the performance of behavioral cloning on confounded Atari environments [5, 12], outperforming various other regularization methods [13, 15, 55, 50] and causality-based methods [12, 47].
- We show that OREO even outperforms inverse reinforcement learning methods trained with a considerable amount of environment interaction [8, 20].

## 2   Related work

**Imitation learning.**   Imitation learning (IL) aims to solve complex tasks where learning a policy from scratch is difficult or even impossible, by learning useful skills or behaviors from expert demonstrations [2, 18, 25, 37, 39, 40, 56]. There are two main approaches for IL: inverse reinforcement learning (IRL) methods that find a cost function under which the expert is uniquely optimal [8, 20, 34, 44, 59], and behavioral cloning methods that formulate the IL problem as a supervised learning problem that predicts expert actions from states [3, 4, 7, 31, 33, 40]. Our work employs

---

[3]de Haan et al. [12] showed that causal discovery methods that depend on faithfulness condition [38] are not applicable to imitation learning setup, as the condition does not hold in environments with nuisance correlates.

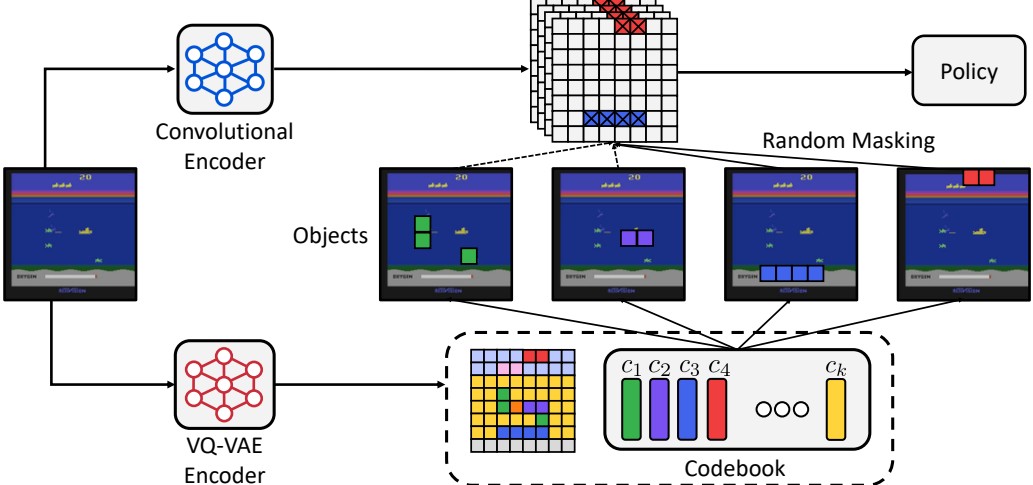

Figure 2: Overview of OREO. We first train a VQ-VAE model that encodes images into discrete codes from a codebook, where each discrete (prototype) representation represents different semantic objects in images. We then regularize a policy by randomly dropping units that share the same discrete code together, i.e., random objects, throughout training.

behavioral cloning as it exhibits the benefit of avoiding costly and dangerous environment interaction, which is crucial for applying imitation learning to real-world scenarios.

**Distributional shift and the causal confusion problem.** Despite its simplicity, BC is known to suffer from the distributional shift, where the state distribution induced by a policy gets different from the training distribution on which the policy was trained. Several approaches have been proposed for learning the policy robust to distributional shift, including interactive IL methods that query experts [42, 43, 51], and regularization techniques [4, 7]. Recently, it has been evidenced that distributional shift leads to the causal confusion problem [4, 12, 54] where a policy exploits the nuisance correlates in states for predicting expert actions. To address this problem, Bansal et al. [4] proposed to randomly drop previous samples from a sequence of samples, and Wen et al. [54] proposed an adversarial training scheme of removing information related to previous actions. The work closest to ours is de Haan et al. [12], which learns a policy with randomly masked disentangled representations and infers the best mask through during environment interaction. Our approach differs in that we regularize the policy to be robust to the causal confusion problem, without any environment interaction.

**Causal discovery from observational data.** Causal discovery aims to discover causal relations among variables by utilizing observational data [38]. Most prior approaches assume that inputs are structured as disentangled variables [6, 16, 26, 36, 48, 47], which often does not hold in domains with high-dimensional inputs, i.e., images. While Lopez-Paz et al. [29] demonstrated the possibility of observational causal discovery from high-dimensional images, combining causal models and representation learning in such domains still remains an open problem [46]. Hence, we instead explore the approach of regularizing a policy that operates on high-dimensional states.

## 3 Method

### 3.1 Preliminaries

We consider the standard imitation learning (IL) framework where an agent learns to solve a target task from expert demonstrations. Specifically, IL is typically defined in the context of a discrete-time Markov decision process (MDP) [52] without an explicitly-defined reward function, which is defined as a tuple $(\mathcal{S}, \mathcal{A}, p, \gamma, \rho_0)$. Here, $\mathcal{S}$ is the state space, $\mathcal{A}$ is the action space, $p(s'|s, a)$ is the transition dynamics, $\rho_0$ is the initial state distribution, and $\gamma \in [0, 1)$ is the discount factor. The goal of IL is to learn a policy $\pi$, mapping from states to actions, using a set of expert demonstrations $\mathcal{D} = \{(s_i, a_i)\}_{i=1}^N$. In our problem setup, an agent cannot interact with the environment, hence it should learn the policy by using only expert demonstrations.

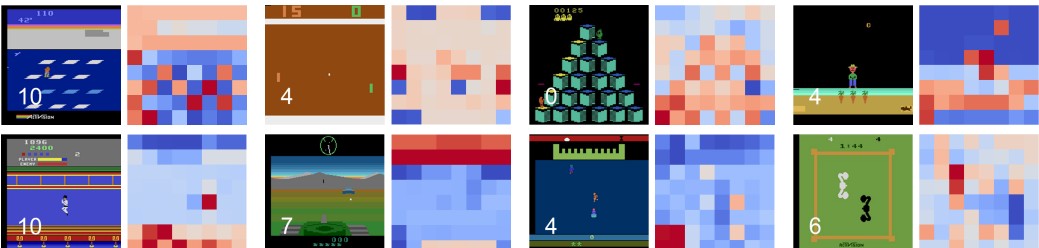

Figure 3: Visualization of the discrete codes from a VQ-VAE model trained on 8 confounded Atari environments, where previous actions are augmented to the images as nuisance variables following the setup in de Haan et al. [12]. The considered environments are Frostbite, Pong, Qbert, Gopher, KungFuMaster, BattleZone, Krull, and Boxing (from left to right, top to bottom). The odd columns show images from environments, and even columns show the corresponding quantized feature maps, respectively. The discrete codes are visualized in 1D using t-SNE [30]. We observe that the units with similar semantics (e.g., the paddles in Pong environment and the carrots in Gopher environment) exhibit similar colors, i.e., mapped into the same or similar discrete codes.

**Behavioral cloning.** Behavioral cloning (BC) reduces an imitation learning problem to the supervised learning problem of training a policy that imitates expert actions. Specifically, we introduce a policy $\pi$ that maps a state $s_t$ to an action $a_t$, and a convolutional encoder $f$ that maps a state $s_t$ to a low-dimensional feature map. Then $\pi$ and $f$ are learned by minimizing the negative log-likelihood of expert actions from demonstrations as follows:

$$\mathcal{L}_{\texttt{BC}}(s_t, a_t) = -\log \pi(a_t | f(s_t)), \tag{1}$$

where $\pi$ is modeled as a multinomial distribution over actions to handle discrete action spaces.

**Vector quantized variational autoencoder.** The VQ-VAE [35] model consists of an encoder $g$ that compresses images into discrete latent representations, and a decoder $d$ that reconstructs images from these discrete representations. Both encoder and decoder share a codebook $C$ of prototype vectors which are also learned throughout training. Formally, given a state $s_t$, the encoder $g$ encodes $s_t$ into a feature map $h_t \in \mathbb{R}^{L \times D}$ that consists of a series of $L$ latent vectors $h_{t,i} \in \mathbb{R}^D, i \in \{1, 2, ..., L\}$. Then $h_t = g(s_t)$ is quantized to discrete representations $e \in \mathbb{R}^{L \times D}$ based on the distance of latent vectors $h_{t,i}$ to the prototype vectors in the codebook $C = \{e_k\}_{k=1}^K$ as follows:

$$e_t = (e_{q(t,1)}, e_{q(t,2)}, \cdots, e_{q(t,L)}), \quad \text{where } q(t,i) = \operatorname*{argmin}_{k \in [K]} \|h_{t,i} - e_k\|_2, \tag{2}$$

where $[K]$ is the set $\{1, \cdots, K\}$. Then the decoder $d$ learns to reconstruct $s_t$ from discrete representations $e_t$. The VQ-VAE is trained by minimizing the following objective:

$$\mathcal{L}_{\texttt{VQVAE}}(s_t) = \underbrace{\|s_t - d(e_t)\|_2^2}_{\mathcal{L}_{\texttt{recon}}} + \underbrace{\|sg\,[h_t] - e_t\|_2^2}_{\mathcal{L}_{\texttt{codebook}}} + \underbrace{\beta \cdot \|sg\,[e_t] - h_t\|_2^2}_{\mathcal{L}_{\texttt{commit}}}, \tag{3}$$

where the operator $sg$ refers to a stop-gradient operator, $\mathcal{L}_{\texttt{recon}}$ is a reconstruction loss for learning representations useful for reconstructing images, $\mathcal{L}_{\texttt{codebook}}$ is a codebook loss to bring codebook representations closer to corresponding encoder outputs $h$, and $\mathcal{L}_{\texttt{commit}}$ is a commitment loss weighted by $\beta$ to prevent encoder outputs from fluctuating frequently between different representations.

## 3.2   OREO: Object-aware regularization for behavioral cloning

In this section, we present OREO: **O**bject-aware **RE**gularizati**O**n that regularizes a policy in an object-aware manner to address the causal confusion problem. Our main idea is to encourage the policy to uniformly attend to all semantic objects in images, in order to prevent the policy from exploiting nuisance variables strongly correlated with expert actions. To this end, we introduce a two-stage approach: we first train a VQ-VAE model that encodes images into discrete codes, then learn the policy with our regularization scheme of randomly dropping units that share the same discrete codes (see Figure 2 and Algorithm 1 for the overview and pseudocode of OREO, respectively).

---
**Algorithm 1** Object-aware regularization (OREO)

---
Initialize parameters of encoder $g$, decoder $d$, codebook $C$, policy $\pi$.
**while** not converged **do**                 // VQ-VAE TRAINING
    Sample $\{s_i\}_{i=1}^{B} \sim \mathcal{D}$.
    Update parameters of $g$, $d$, $C$ by minimizing $\sum_{i=1}^{B} \mathcal{L}_{\texttt{VQVAE}}(s_i)$
**end while**
Initialize encoder $f$ with parameters of $g$.
**while** not converged **do**            // UPDATE POLICY VIA BEHAVIORAL CLONING
    Sample $\{s_i, a_i\}_{i=1}^{B} \sim \mathcal{D}$
    Get random masks $\{M_i\}_{i=1}^{B}$ in (4)
    Update parameters of $f$, $\pi$ by minimizing $\sum_{i=1}^{B} \mathcal{L}_{\texttt{OREO}}(s_i, a_i, M_i)$ in (5)
**end while**

---

**Extracting semantic objects.** To regularize a policy in an object-aware manner, we propose to utilize discrete representations from a VQ-VAE model trained by optimizing the objective in (3) with images from expert demonstrations $\mathcal{D}$. Our motivation comes from the observation that the units of a feature map corresponding to similar objects are mapped into similar discrete codes (see Figure 3). Then, in order to extract semantic objects from images and utilize them for regularizing the policy, we propose to randomly drop the units of a feature map that share the same discrete code together throughout training. Formally, for each state $s_t$, we sample $K$ binary random variables $m_k \in \{0, 1\}$, $k = 1, 2, ..., K$ from a Bernoulli distribution with probability $1 - p$, where $p$ is the drop probability. Then, we construct a mask $M_t$ by utilizing the discrete representations $e_t$ in (2) as follows:

$$M_t = (m_{q(t,1)}, m_{q(t,2)}, \cdots, m_{q(t,L)}), \quad \text{where } q(t, i) = \underset{k \in [K]}{\operatorname{argmin}} \|h_{t,i} - e_k\|_2. \tag{4}$$

By considering units of a feature map with the same discrete code, we remark that our method can effectively extract semantic objects from high-dimensional images.

**Behavioral cloning with OREO.** Now we propose to utilize our object-aware masking scheme for the regularization of a policy. To this end, we first initialize a convolutional encoder $f$ with the parameters of a VQ-VAE encoder $g$. We empirically find that employing $f$ as our backbone encoder for $\pi$ instead of a fixed encoder $g$ is more effective, as it allows an encoder to learn useful information for predicting actions. Then, we train the policy $\pi$ by minimizing the following objective:

$$\mathcal{L}_{\texttt{OREO}}(s_t, a_t, M_t) = -\log \pi(a_t | f(s_t) \odot M_t), \tag{5}$$

where $\odot$ denotes elementwise product, and the mask $M_t$ is shared across all channels in a feature map $f(s_t)$. Here, our intuition is that our object-aware regularization scheme should be useful for enforcing the policy not to exploit specific objects strongly correlated with expert actions, as the policy should utilize all semantic objects throughout training. Additionally, following Srivastava et al. [50], we scale the masked features by a factor of $1/(1 - p)$ during the training to ensure the scale of the expected output with masked features to match the scale of outputs at test time.

## 4 Experiments

In this section, we designed our experiments to answer the following questions:

- How does OREO compare to other regularization schemes that randomly drop units from a feature map [15, 50], data augmentation schemes [13, 55], and causality-based methods [12, 47] (see Table 1)?

- How does OREO compare to inverse reinforcement learning methods that learn a policy with environment interaction [8, 20] (see Figure 5)?

- Why is regularization necessary for addressing the causal confusion problem (see Figure 6a), and why is OREO effective for addressing this problem (see Figure 8)?

- Can OREO improve BC using various sizes of expert demonstrations (see Figure 7)?

- Can OREO also address the causal confusion problem when inputs are high-dimensional, complex real-world images (see Table 3)?

Table 1: Performance of policies trained on various confounded Atari environments without environment interaction. OREO achieves the best score on 15 out of 27 environments, and the best median and mean human-normalized score (HNS) over all environments. The results for each environment report the mean of returns averaged over eight runs. We provide standard deviations in Appendix I. CCIL$^\dagger$ denotes the results without environment interaction.

| Environment | BC | Dropout | DropBlock | Cutout | RandomShift | CCIL$^\dagger$ | CRLR | OREO |
|---|---|---|---|---|---|---|---|---|
| Alien | 954.1 | 1003.8 | 926.4 | 973.3 | 806.5 | 820.0 | 82.5 | **1056.2** |
| Amidar | 95.8 | 89.4 | 110.1 | **118.7** | 98.0 | 74.9 | 12.0 | 105.7 |
| Assault | 793.8 | 820.4 | 815.0 | 687.6 | 828.9 | 683.3 | 0.0 | **840.9** |
| Asterix | 292.2 | 313.8 | 345.4 | 212.4 | 135.5 | 643.2 | **650.0** | 180.8 |
| BankHeist | 442.1 | 485.7 | 508.4 | 486.1 | 367.2 | **653.5** | 0.0 | 493.9 |
| BattleZone | 11921.2 | 12457.5 | 12025.0 | 11107.5 | 9180.0 | 6370.0 | 1468.8 | **12700.0** |
| Boxing | 18.8 | 20.3 | 32.2 | 20.5 | **38.3** | 34.8 | -43.0 | 36.4 |
| Breakout | **5.7** | 5.4 | 4.8 | 1.0 | 2.0 | 0.5 | 0.0 | 4.2 |
| ChopperCommand | 874.2 | 921.4 | 919.4 | 1016.1 | 936.4 | 760.6 | **1077.2** | 977.4 |
| CrazyClimber | 45372.9 | 39501.6 | 38345.6 | 44523.2 | 41924.0 | 22616.8 | 112.5 | **55523.4** |
| DemonAttack | 157.2 | 180.5 | 167.8 | 173.1 | **241.8** | 171.3 | 0.0 | 224.5 |
| Enduro | 241.4 | 250.4 | 341.8 | 119.6 | 316.4 | 143.1 | 3.9 | **522.8** |
| Freeway | 32.3 | 32.4 | 32.7 | 32.5 | 33.0 | **33.1** | 21.4 | 32.7 |
| Frostbite | 116.3 | 124.5 | 128.2 | **139.4** | 121.6 | 53.3 | 80.0 | 129.9 |
| Gopher | 1713.9 | 1819.1 | 1818.2 | 1481.0 | 1995.0 | 1404.5 | 0.0 | **2515.0** |
| Hero | 11923.1 | 14109.7 | 14711.4 | 14896.6 | 12816.0 | 6567.8 | 346.2 | **15219.8** |
| Jamesbond | 419.0 | 451.0 | 473.8 | 381.8 | 428.4 | 387.2 | 0.0 | **502.8** |
| Kangaroo | 2781.5 | 2912.9 | 3217.1 | 2824.0 | 1923.9 | 1670.5 | 122.8 | **3700.2** |
| Krull | 3634.3 | 3892.1 | 3832.1 | 3656.4 | 3788.7 | 3090.8 | 0.1 | **4051.6** |
| KungFuMaster | 15074.8 | 14452.1 | 15753.0 | 11405.6 | 13389.9 | 13394.9 | 0.0 | **18065.6** |
| MsPacman | 1432.9 | 1733.1 | 1446.4 | 1711.0 | 1223.5 | 1084.2 | 105.3 | **1898.4** |
| Pong | 3.2 | 10.2 | 11.5 | 6.8 | -0.1 | -2.7 | -21.0 | **14.2** |
| PrivateEye | 2681.8 | 2599.1 | 2720.6 | 2670.6 | **3969.2** | 305.3 | -1000.0 | 3124.9 |
| Qbert | 5438.4 | 6469.0 | 6140.3 | 5748.6 | 3921.4 | 5138.0 | 125.0 | **6966.4** |
| RoadRunner | 18381.5 | 21470.9 | 22265.4 | 12417.1 | 16210.0 | 11834.1 | 1022.9 | **24644.2** |
| Seaquest | 454.4 | 471.3 | 486.8 | 330.1 | **1016.8** | 271.2 | 172.5 | 753.1 |
| UpNDown | 4221.1 | 4147.1 | **4789.2** | 4159.6 | 3880.2 | 2631.1 | 20.0 | 4577.9 |
| Median HNS | 44.1% | 47.4% | 49.8% | 42.0% | 47.6% | 36.2% | -1.5% | **51.2%** |
| Mean HNS | 73.2% | 79.0% | 91.7% | 69.5% | 88.1% | 71.7% | -45.9% | **105.6%** |

**Environments and datasets.** We evaluate OREO on 27 Atari environments [5], which are selected by following prior works [12, 49]. Following de Haan et al. [12], we consider *confounded* Atari environments, where images are augmented with previous actions (see Figure 4). We utilize a single frame as an input to a policy, to focus on the causal confusion problem from nuisance correlates in the current state[4]. In our experiments, we report two evaluation metrics: average score

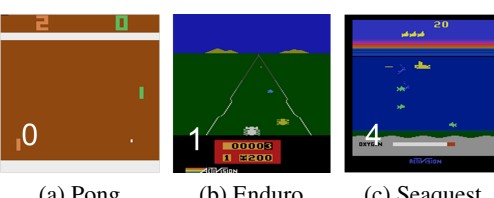

(a) Pong    (b) Enduro    (c) Seaquest

Figure 4: Confounded Atari environments with previous actions (white number in lower left).

from environments and human-normalized score (HNS := $\frac{\text{Agent}_{\text{score}}-\text{Random}_{\text{score}}}{\text{Human}_{\text{score}}-\text{Random}_{\text{score}}}$), following Mnih et al. [32]. For expert demonstrations, we utilize DQN Replay dataset [1]. As this dataset consists of 50M transitions of each environment collected during the training of a DQN agent [32], we use the last $N$ trajectories as expert demonstrations. We preprocess input images to grayscale images of $84 \times 84 \times 1$, by utilizing Dopamine library [9]. We provide more details in Appendix B.

**Implementation.** We use a single Nvidia P100 GPU and 8 CPU cores for each training run. The training time for OREO is 6 hours on the dataset of size 50000, compared to 3 hours for BC, which is because OREO additionally trains a VQ-VAE model. As for hyperparameter selection, we use the default hyperparameters from previous or similar works [35, 50], i.e., a drop probability of $p = 0.5$, a codebook size of $K = 512$, and a commitment cost of $\beta = 0.25$. We use the same hyperparameters across all environments. We report the results over 8 runs unless specified. Source code and more details on implementation are available in Appendix A and B, respectively.

---

[4]We refer to Wen et al. [54] for the discussion on the causal confusion problem from stacking states. While we mainly focus on the single frame setup, OREO is also effective on multiple frame setup (see Appendix F).

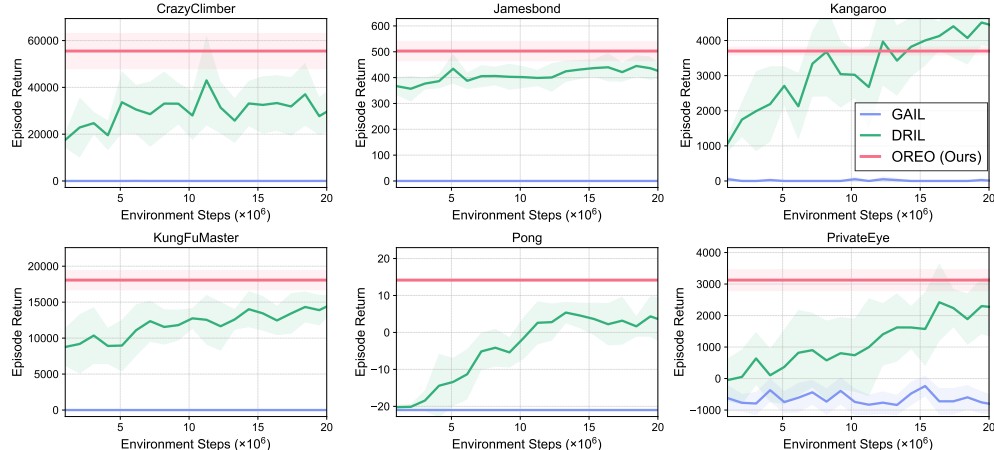

Figure 5: We compare OREO to inverse reinforcement learning methods that require environment interaction for learning a policy, on 6 confounded Atari environments. OREO outperforms baseline methods in most cases, even without using any interaction with environments. The solid line and shaded regions represent the mean and standard deviation, respectively, across eight runs.

**Baselines.** We consider BC as the most basic baseline method. To evaluate the effectiveness of our object-aware regularization scheme, we compare to regularization techniques that drop the randomly sampled units (i.e., Dropout [50]) or randomly sampled blocks (i.e., DropBlock [15]) from the feature map of a convolutional encoder. We also compare to data augmentation schemes, i.e., Cutout [13] that randomly masks out a square patch from images, and RandomShift [55] that randomly shifts pixels of images for regularization. We also consider the method of de Haan et al. [12] that learns a policy on top of disentangled representations from a $\beta$-VAE [19] (i.e., CCIL), and an observational causal inference method that estimates the causal contribution of each variable by confounder balancing (i.e., CRLR [47]). We provide the details for baselines in Appendix B and H.

**Comparative evaluation.** Table 1 shows the performance of various methods that learn a policy without environment interaction. OREO significantly improves the performance of BC in most environments, outperforming other regularization techniques. In particular, OREO achieves the mean HNS of 105.6%, while the second-best method, i.e., DropBlock, achieves 91.7%. This demonstrates that our object-aware regularization scheme is indeed effective for addressing the causal confusion problem (see Figure 8 for qualitative experimental results). We found that CCIL without environment interaction does not exhibit strong performance in most environments, possibly due to the difficulty of learning disentangled representations from high-dimensional images [28]. We also provide experimental results for CCIL with environment interaction in Appendix D, where the performance slightly improves but overall trends are similar. We observe that CRLR underperforms in most environments, which shows the difficulty of causal inference from high-dimensional images. We emphasize that OREO also outperforms other baselines in original Atari environments, which implies that our method is also effective for addressing the causal confusion that naturally occurs (see Figure 1). We provide experimental results for the original setup in Appendix C.

**Comparison with inverse reinforcement learning methods.** To demonstrate that OREO can exhibit strong performance without environment interaction, we compare our method to inverse reinforcement learning (IRL) methods that first learn a reward function using expert demonstrations, and train a policy with environment interaction using learned reward function. Specifically, we consider GAIL [20], a method that learns reward function by discriminating expert states from on-policy states during environment interaction; and DRIL [8], one of the strongest IRL methods that utilizes the disagreement between ensemble policies as a reward function. As shown in Figure 5, we observe that OREO exhibits superior performance to GAIL and DRIL on most confounded Atari environments, which are trained with 20M environment steps following the setup in Brantley et al. [8]. While IRL methods might outperform OREO asymptotically with more environment interaction, this result demonstrates that OREO indeed allows for achieving strong performance without interaction. We also found that GAIL exhibits almost zero performance in most environments, which is similar to the observation of previous works that GAIL suffers in environments with high-dimensional image inputs [8, 12, 41]. We remark that OREO can also be applied to IRL methods (see Appendix E for relevant experimental results of DRIL + OREO on confounded Atari environments).

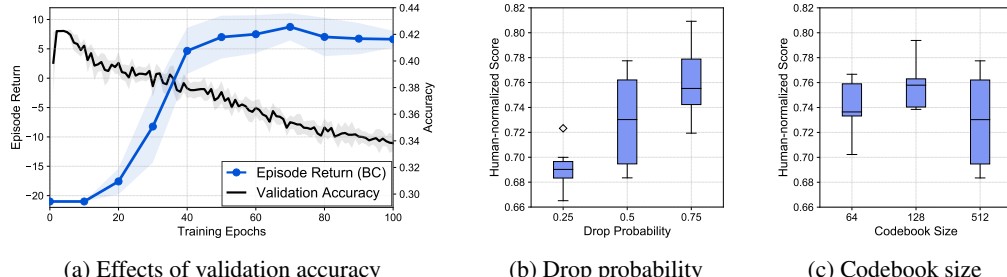

| (a) Effects of validation accuracy | (b) Drop probability | (c) Codebook size |

Figure 6: (a) Score and validation accuracy on confounded Pong environment, where validation accuracy is not aligned with the score at test time, which necessitates the use of regularization for addressing the causal confusion problem. We visualize the performance of OREO over 8 confounded Atari environments with varying (b) the drop probability of each code from a codebook and (c) codebook size. Boxplots are drawn using mean human-normalized scores obtained from eight runs.

**Why is regularization necessary in confounded environments?** A simple and widely used approach to address the overfitting problem in supervised learning is a model selection with a validation dataset. To see how this works in our setup, we first introduce a validation dataset consisting of 5 expert demonstrations on confounded Pong environment, and visualize the scores and validation accuracies measured by a policy learned with BC in Figure 6a. We observe that this simple scheme is not helpful for confounded Atari environments, i.e., validation accuracy is not aligned with the score at test time, because the distribution of the validation dataset could be significantly different from the distribution induced by a learned policy. As the evaluation of the policy in environments during training could be dangerous or even impossible, this result implies that regularizing the policy is necessary for successful imitation learning in confounded environments.

**Effects of hyperparameters.** We investigate the effect of two major hyperparameters, i.e., $p \in \{0.25, 0.5, 0.75\}$ for the drop probability in (4), and $K \in \{64, 128, 512\}$ for the codebook size in (2). Figure 6b shows that the performance improves as $p$ increases, which implies that more strong regularization is effective for addressing the causal confusion problem on confounded Atari environments. Figure 6c shows that too small or large codebook size could be harmful to the performance. We remark that our experiments used default hyperparameters $p = 0.5$ and $K = 512$ for reporting the results, so the performance of OREO could be further improved with more tuning.

**Effects of expert demonstration size.** To investigate the effectiveness of OREO with various sizes of expert demonstrations, we evaluate the performance of OREO with a varying number of expert demonstrations $N \in \{5, 10, 20, 35, 50\}$. Specifically, we report the mean HNS over 8 confounded Atari environments, which are randomly selected due to the high computation cost of running experiments for all environments. As shown in Figure 7, OREO consistently improves the performance of BC across a wide range of dataset sizes. As for the comparison with other baselines, OREO achieves superior performance to Dropout and DropBlock except for the extreme case of $N = 5$, which is because learning a VQ-VAE model with a limited number of data could be unstable. We also observe that DropBlock and OREO consistently outperform Dropout, which supports our intuition that dropping individual units from a feature map is not sufficient for effective regularization to address the causal confusion problem.

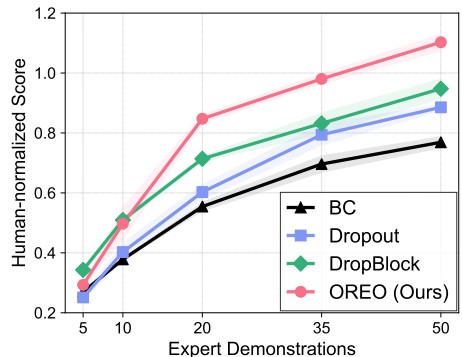

Figure 7: Mean human-normalized score over 8 confounded Atari environments with a varying number of expert demonstrations. The solid line and shaded regions represent the mean and standard deviation, respectively, across four runs.

Table 2: Performance of policies trained on various confounded Atari environments without environment interaction. VQ-VAE + BC learns a BC policy on top of fixed VQ-VAE representations. The results for each environment report the mean and standard deviation of returns over eight runs.

| Environment | BC | VQ-VAE + BC | VQ-VAE + Dropout | VQ-VAE + DropBlock | OREO |
|---|---|---|---|---|---|
| BankHeist | 442.1$\pm$ 20.7 | 358.8$\pm$ 25.8 | 491.1$\pm$ 28.9 | 488.0$\pm$ 49.7 | **493.9$\pm$ 17.6** |
| Enduro | 241.4$\pm$ 28.4 | 154.6$\pm$ 10.7 | 57.1$\pm$ 12.6 | 111.2$\pm$ 16.4 | **522.8$\pm$ 29.1** |
| KungFuMaster | 15074.8$\pm$ 275.5 | 11055.1$\pm$ 867.2 | 13323.0$\pm$ 1390.0 | 14861.1$\pm$ 1561.5 | **18065.6$\pm$ 1411.5** |
| Pong | 3.2$\pm$ 0.7 | 3.6$\pm$ 1.8 | 10.4$\pm$ 0.8 | 13.6$\pm$ 0.3 | **14.2$\pm$ 0.4** |
| PrivateEye | 2681.8$\pm$ 270.2 | 2255.8$\pm$ 569.5 | 390.2$\pm$ 300.9 | 746.8$\pm$ 527.8 | **3124.9$\pm$ 349.6** |
| RoadRunner | 18381.5$\pm$ 1519.9 | 5783.2$\pm$ 403.6 | 6633.8$\pm$ 716.8 | 7771.1$\pm$ 843.6 | **24644.2$\pm$ 2235.1** |
| Seaquest | 454.4$\pm$ 53.5 | 344.9$\pm$ 35.2 | 325.6$\pm$ 28.2 | 396.6$\pm$ 36.8 | **753.1$\pm$ 63.6** |
| UpNDown | 4221.1$\pm$ 214.5 | 2676.9$\pm$ 268.9 | 3310.8$\pm$ 536.2 | 4073.9$\pm$ 760.9 | **4577.9$\pm$ 307.6** |
| Median HNS | 62.7% | 47.9% | 45.3% | 53.2% | **72.9%** |
| Mean HNS | 70.8% | 41.3% | 45.7% | 53.0% | **100.1%** |

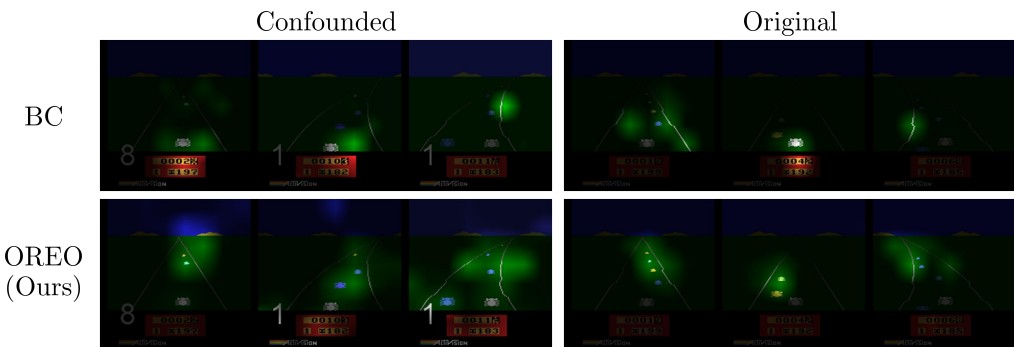

Figure 8: We visualize the spatial attention map from a convolutional encoder trained with BC and OREO in confounded (left) and original (right) Enduro environment. We observe that the encoder trained with OREO attends to important objects from images (e.g., approaching cars), while the encoder trained with BC only attends the small region around a car.

**Contribution of a separate convolutional encoder.** In order to verify the effect of introducing an additional convolutional encoder $f$ instead of learning a policy on top of the fixed VQ-VAE encoder $g$ in (5), we provide the experimental results that evaluate VQ-VAE + BC, where a BC policy is learned on top of fixed VQ-VAE representations in Table 2. We first observe that the performance of VQ-VAE + BC performs worse than vanilla BC, which is because fixed VQ-VAE representations learned by reconstruction objective (3) do not contain fine-grained features required for imitating expert actions. Instead, one can see that OREO significantly improves the performance of BC and outperforms all baselines based on fixed VQ-VAE representations, achieving the mean HNS of 100.1% compared to 53.0% of VQ-VAE + DropBlock. This shows that OREO is not the naïve combination of VQ-VAE and BC, but a carefully designed method to exploit the discrete codes from VQ-VAE for object-aware regularization to address the causal confusion problem.

**How does OREO improve the performance of BC?** To understand how OREO improves the performance of BC, we visualize spatial attention maps from the last convolutional layer of a policy encoder in Figure 8. Specifically, following prior works [24, 57], we compute a spatial attention map by averaging the absolute values of a feature map along the channel dimension. We then apply 2-dimensional spatial softmax and multiply the upscaled attention map with images for visualization. We observe that the activations from the encoder trained with OREO are capturing all important objects of the environment (e.g., car that an agent controls and approaching cars), while the activations from BC are missing information of approaching cars by only focusing on the small region around a car and a scoreboard. This shows that our regularization scheme that encourages a policy to uniformly attend to all semantic objects allows for learning the policy that attends to important objects.

**Effectiveness on real-world applications.** To further demonstrate the effectiveness of OREO on real-world applications where inputs are high-dimensional, complex images, we additionally consider a self-driving CARLA environment [14]. Specifically, we train a conditional imitation learning policy

Table 3: Performance of policies trained on 150 expert demonstrations from the CARLA driving dataset, under a weather condition of daytime. The results for each environment report the mean and standard deviation of success rates over four runs. OREO achieves the best success rate on all tasks.

| Task | BC | Dropout | DropBlock | OREO |
|------|-----|---------|-----------|------|
| Straight | $75.0\pm_{1.7}$ | $82.0\pm_{8.3}$ | $74.0\pm_{3.5}$ | $\textbf{87.0}\pm_{\textbf{4.4}}$ |
| One turn | $43.0\pm_{9.1}$ | $59.0\pm_{3.3}$ | $53.0\pm_{5.2}$ | $\textbf{70.0}\pm_{\textbf{7.2}}$ |
| Navigation | $16.9\pm_{7.6}$ | $30.4\pm_{10.7}$ | $21.7\pm_{9.2}$ | $\textbf{35.7}\pm_{\textbf{10.2}}$ |
| Navigation w/ dynamic obstacles | $18.0\pm_{4.5}$ | $26.0\pm_{6.0}$ | $19.0\pm_{5.2}$ | $\textbf{30.0}\pm_{\textbf{4.5}}$ |

[10] using 150 expert demonstrations from the dataset [53] consisting of $200 \times 88 \times 3$ real-world images under a weather condition of daytime. Table 3 shows the average success rate of OREO and baseline methods on four CARLA benchmark tasks, i.e., Straight, One turn, Navigation, and Navigation with dynamics obstacles, where each task consists of 25 different navigation routes. The results show that OREO improves the performance of BC and outperforms other regularization methods, which implies that our object-aware regularization can also be effective on more complex real-world applications.

## 5 Discussion

In this paper, we present OREO, a simple regularization method to address the causal confusion problem in imitation learning. OREO regularizes a policy in an object-aware manner, by randomly dropping the units of a feature map that share the same discrete codes from a VQ-VAE model. Our experimental results demonstrate that OREO improves the performance of behavioral cloning without costly environment interaction, which is crucial for safe and successful imitation learning.

**Limitations.** One limitation of our method is that it is only designed to regularize a policy when inputs are images, and not applicable to state-based environments. However, we still believe that OREO can be a practical solution to the causal confusion problem in various image-based applications, e.g., video games [32], self-driving [7], and robotic manipulation [23]. Another limitation is that we do not deduce the cause-effect relations for addressing the causal confusion problem, but instead regularize the policy to prevent it from exploiting nuisance correlates. However, given that it is an open problem to infer the structured disentangled variables and discover the causal relations among the variables [46], we believe encouraging the policy to attend to *all* semantic objects is a reasonable and promising direction for addressing this problem.

**Potential negative impacts.** Real-world applications of behavioral cloning, e.g., autonomous driving [4], require a large amount of data that often contain sensitive information, therefore raising privacy concerns. As our method is built upon a variational autoencoder, it could be exposed to privacy violation attacks that infer training data information, such as model inversion [58], and membership inference [17]. For example, the facial information of pedestrians may be reconstructed via membership inference attack. To address this vulnerability to privacy violation attacks, a differentially private variational autoencoder would be required for real applications. In addition, pre-training VQ-VAE requires additional computing resources, which might lead to the increased energy cost for learning imitation learning policies. Also, a behavioral cloning policy will imitate whatever demonstrations one specifies. If some bad actions are included in expert demonstrations, the policy would perform dangerous actions to users. For these reasons, in addition to developing algorithms for better performance, it is also important to consider safe adaptation.

## Acknowledgments and Disclosure of Funding

This research was supported by Microsoft Research Asia, Institute of Information & communications Technology Planning & Evaluation (IITP) grant funded by the Korea government (MSIT) (No.2019-0-00075, Artificial Intelligence Graduate School Program(KAIST)), and the MSIT (Ministry of Science, ICT), Korea, under the High-Potential Individuals Global Training Program) (No.2020-0-01649) supervised by the IITP (Institute for Information & Communications Technology Planning &

Evaluation. We would like to thank Kimin Lee, Sangwoo Mo, Seonghyeon Park, Sihyun Yu, and anonymous reviewers for providing helpful feedbacks and suggestions in improving our paper.

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
