# Appendix

## A    Source codes

Source codes for reproducing our experimental results are available at `https://github.com/alinlab/oreo`.

## B    Details on Atari experiments

### B.1    Experimental setup

**Environments and datasets.**    We utilize DQN Replay dataset[5] [1] for expert demonstrations on 27 Atari environments [5]. To encourage the size of the dataset to be consistent across multiple environments, we use the number of expert demonstrations $N \in \{20, 50\}$. We provide the size of a dataset for each environment in Table 4. We process input images to grayscale images of $84 \times 84 \times 1$, by utilizing Dopamine library[6] [9]. Following de Haan et al. [12], we consider *confounded* Atari environments, where images are augmented with previous actions (see Figure 4). We provide source codes for loading images from the dataset, preprocessing images, and augmenting numbers to the images in Section A. For experiments with selected environments in Figure 7, we randomly chose 8 confounded Atari environments, i.e., BankHeist, Enduro, KungFuMaster, Pong, PrivateEye, RoadRunner, Seaquest, and UpNDown, due to the high computational cost of considering all environments.

**Evaluation.**    (a) For all experimental results without environment interaction, we train a policy for 1000 epochs without early stopping based on validation accuracy (see Figure 6a for how early stopping is not effective in our setup), and report the final performance of the trained policy. Specifically, we average the scores over 100 episodes evaluated on confounded environments for each random seed. (b) For all experimental results with inverse reinforcement learning methods that require environment interaction (i.e., GAIL [20] and DRIL [8]), we evaluate a policy over 10 episodes every 1M environment step during the training.

Table 4: Dataset size of each Atari environment.

| Environment | $N$ | Data | Environment | $N$ | Data | Environment | $N$ | Data |
|---|---|---|---|---|---|---|---|---|
| Alien | 50 | 53165 | CrazyClimber | 20 | 83557 | Krull | 50 | 65701 |
| Amidar | 20 | 57155 | DemonAttack | 20 | 47727 | KungFuMaster | 20 | 57235 |
| Assault | 50 | 58868 | Enduro | 20 | 169767 | MsPacman | 50 | 64305 |
| Asterix | 20 | 68126 | Freeway | 20 | 41020 | Pong | 20 | 41402 |
| BankHeist | 50 | 58516 | Frostbite | 50 | 24043 | PrivateEye | 20 | 54020 |
| BattleZone | 50 | 83061 | Gopher | 20 | 44011 | Qbert | 50 | 59379 |
| Boxing | 50 | 47170 | Hero | 50 | 68903 | RoadRunner | 50 | 60546 |
| Breakout | 50 | 63799 | Jamesbond | 20 | 33659 | Seaquest | 20 | 37682 |
| ChopperCommand | 50 | 35262 | Kangaroo | 20 | 45898 | UpNDown | 50 | 74348 |

### B.2    Implementation details

**Implementation details for OREO.**

- **VQ-VAE training.** We use the publicly available implementation of VQ-VAE[7] modified to make it work with images of size $84 \times 84 \times 1$. Specifically, The encoder consists of four convolutional layers, three with stride 2 and kernel size $4 \times 4$ and one with stride 1 and kernel size $3 \times 3$, followed by 2 residual $3 \times 3$ blocks (implemented as ReLU, $3 \times 3$ Conv, ReLU, $1 \times 1$ conv), all having 256 hidden units. The decoder similarly has 2 residual $3 \times 3$ blocks, followed by four transposed convolution layers, one with stride 1 and kernel size $3 \times 3$, and three with stride 2

---

[5]`https://research.google/tools/datasets/dqn-replay`
[6]`https://github.com/google/dopamine`
[7]`https://github.com/zalandoresearch/pytorch-vq-vae`

and kernel size $4 \times 4$. For training, we train a VQ-VAE model for 1000 epochs with a batch size of 1024. We use Adam optimizer with the learning rate of 3e-4. As for the hyperparameters of VQ-VAE, we use a codebook size of $K = 512$, and a commitment cost of $\beta = 0.25$, following the original implementation.

- **Behavioral cloning with OREO.** For efficient implementation of OREO, we first compute quantized discrete codes of all images in datasets with pre-trained VQ-VAE, instead of processing every image through VQ-VAE encoder during the training. Then we utilize stored discrete codes for obtaining random masks for training a policy. We find that generating multiple random masks for each image and aggregating the loss computed with each mask marginally improves the performance, by providing more diverse features to the policy. In our experiments, we generate 5 random masks during training. We train a policy for 1000 epochs with the batch size of 1024, and use Adam optimizer with the learning rate of 3e-4.

**Implementation details for regularization and causality-based methods.**

- **Behavioral cloning.** We train a BC policy by optimizing the objective in (1) using states and actions from expert demonstrations. Note that other regularization baselines are based on BC.

- **Dropout.** Dropout [50] is a regularization technique that drops units of a feature map from a convolutional encoder. Specifically, for all units of a feature map, Dropout samples binary random variables from a Bernoulli distribution with probability $1 - p$, and apply the randomly sampled masks throughout training. We use `nn.Dropout` from PyTorch[8] library with $p = 0.5$.

- **DropBlock.** DropBlock [15] is a regularization technique that drops units in a contiguous region of a feature map, i.e., blocks, with the default hyperparameters of $p = 0.3$ and the block size of 3. We use the publicly available implementation of DropBlock[9] for our experiments. Following this original implementation, we linearly increase $p$ from 0 to the target value during training.

- **Cutout.** Cutout [13] randomly masks out a square patch from images. We randomly sampled the size of the patch from $10 \times 10$ to $30 \times 30$, by using `RandomErasing` from Kornia[10] library.

- **RandomShift.** RandomShift [55] is a regularization technique that *shifts* images by randomly sampled pixels. Specifically, it pads each side of an image by 4 pixels with boundary pixels and performs random crop of size $84 \times 84$. We implemented RandomShift by following the publicly available implementation[11] from the authors.

- **CCIL.** CCIL (named after Causal Confusion in Imitation Learning; [12]) is an interventional causal discovery method that first (i) learns disentangled representations from $\beta$-VAE [19] and (ii) infers the causal graph during environment interaction. As publicly available implementation[12] only contains source code that works on the low-dimensional MountainCar environment, we faithfully reproduce the method and report the results. Specifically, we employ CoordConv[13] [27] for both the encoder and decoder architectures of $\beta$-VAE. We find that prediction accuracy of a policy trained using a fixed $\beta$-VAE does not improve over chance level accuracy, possibly because a reconstruction task is not sufficient for learning representations that capture the information required for predicting actions. Hence, we additionally introduce an action prediction task when training a $\beta$-VAE, which we find crucial for improving the accuracy over chance level accuracy.

- **CRLR.** As CRLR requires inputs to be binary values, we develop and compare to the categorical version of CRLR that works on top of VQ-VAE discrete codes (see Appendix H).

**Implementation details for inverse reinforcement learning methods.** For all inverse reinforcement learning (IRL) methods, we use the publicly available implementation (https://github.com/xkianteb/dril) for reporting the results, with additional modification to original source code to train and evaluate a policy on confounded Atari environments.

---

[8] https://pytorch.org
[9] https://github.com/miguelvr/dropblock
[10] https://github.com/kornia/kornia
[11] https://github.com/denisyarats/drq
[12] https://github.com/pimdh/causal-confusion
[13] https://github.com/walsvid/CoordConv

- **GAIL.** GAIL [20] is an IRL method that learns a discriminator network that distinguishes expert states from states visited by the current policy, and utilizes the negative output of the discriminator as a reward signal for learning RL agents during environment interaction.
- **DRIL.** DRIL [8] is an IRL method that learns an ensemble of behavioral cloning policies and utilizes the disagreement (i.e., variance) between the predictions of ensemble policies as a cost signal (the negative of reward signal) for learning RL agents during environment interaction.

## C    Comparative evaluation on original Atari environments

Table 5 shows the performance of various methods which do not use environment interaction, on original Atari environments. We observe that OREO significantly improves behavioral cloning, also outperforming baseline methods. In particular, OREO achieves the mean HNS of 114.9%, while the second-best method, i.e., DropBlock, achieves 99.0%. This demonstrates that our object-aware regularization scheme is also effective for addressing the causal confusion that naturally occurs in the dataset (see Figure 1).

Table 5: Performance of policies trained on various original Atari environments without environment interaction. OREO achieves the best score on 14 out of 27 environments, and the best median and mean human-normalized score (HNS) over all environments. The results for each environment report the mean of returns averaged over eight runs. CCIL[†] denotes the results without environment interaction.

| Environment | BC | Dropout | DropBlock | Cutout | RandomShift | CCIL[†] | CRLR | OREO |
|---|---|---|---|---|---|---|---|---|
| Alien | 986.5 | 1117.2 | 1094.8 | 1104.4 | 863.5 | 1050.4 | 100.0 | **1222.2** |
| Amidar | 90.8 | 81.6 | 113.5 | 125.0 | 78.2 | 78.6 | 12.0 | **130.5** |
| Assault | 816.8 | 901.1 | 829.9 | 694.1 | 848.7 | 755.5 | 0.0 | **905.2** |
| Asterix | 249.0 | 176.6 | 252.2 | 195.0 | 99.1 | 314.1 | **592.5** | 212.5 |
| BankHeist | 399.0 | 476.6 | 471.2 | 442.5 | 354.8 | **606.1** | 0.0 | 448.4 |
| BattleZone | 10933.8 | 11621.2 | **12067.5** | 10641.2 | 8748.8 | 11191.2 | 5615.0 | 11703.8 |
| Boxing | 21.8 | 25.7 | 32.1 | 21.2 | 35.8 | 34.2 | -43.0 | **39.9** |
| Breakout | **6.4** | 2.9 | 6.0 | 3.1 | 4.4 | 2.1 | 0.0 | 5.4 |
| ChopperCommand | 1163.0 | 1162.0 | 1161.8 | 1183.9 | 1026.2 | 1027.2 | 1070.2 | **1282.9** |
| CrazyClimber | 54142.2 | 54965.4 | 55854.0 | 47456.4 | 60465.9 | 39015.2 | 885.5 | **69380.1** |
| DemonAttack | 238.8 | **359.3** | 225.6 | 217.8 | 294.8 | 194.6 | 22.7 | 0.0 |
| Enduro | 226.2 | 304.6 | 359.1 | 132.9 | 282.2 | 182.8 | 0.8 | **514.4** |
| Freeway | 32.3 | 32.6 | 32.6 | 32.8 | 33.0 | **33.1** | 21.4 | 32.9 |
| Frostbite | 153.6 | 149.2 | **165.7** | 135.2 | 133.2 | 96.7 | 78.1 | 152.7 |
| Gopher | 1874.4 | 2220.4 | 2040.5 | 1588.2 | 1456.2 | 1301.9 | 0.0 | **2903.9** |
| Hero | 15100.4 | 15994.4 | 17058.6 | 15971.8 | 14867.2 | **17487.6** | 0.0 | 16370.3 |
| Jamesbond | 447.6 | 492.3 | 481.9 | 418.9 | 452.1 | 460.4 | 0.0 | **527.9** |
| Kangaroo | 3162.8 | 2860.4 | **3638.6** | 3242.6 | 2202.1 | 2938.1 | 0.0 | 3602.9 |
| Krull | 4447.9 | **4764.7** | 4526.5 | 4270.6 | 4611.6 | 4247.1 | 0.0 | 4633.6 |
| KungFuMaster | 12900.6 | 14994.5 | 14819.0 | 9956.9 | 11698.0 | 12876.9 | 0.0 | **16955.5** |
| MsPacman | 1921.9 | 2022.6 | 2151.7 | 1949.7 | 1046.3 | 1160.6 | 70.0 | **2263.8** |
| Pong | 3.7 | 10.0 | 11.6 | 7.8 | 0.8 | -19.8 | -21.0 | **12.5** |
| PrivateEye | 3035.4 | 3396.3 | 3057.6 | 3092.2 | **3578.9** | 1016.4 | -1000.0 | 3162.6 |
| Qbert | 5925.4 | **6363.1** | 5904.3 | 6174.8 | 4100.1 | 5056.3 | 125.0 | 5763.4 |
| RoadRunner | 18010.1 | 20137.8 | 22522.5 | 12698.9 | 15615.4 | 18985.2 | 1528.6 | **27303.9** |
| Seaquest | 527.5 | 644.4 | 622.3 | 376.6 | **948.0** | 402.4 | 169.8 | 921.0 |
| UpNDown | 3782.1 | 3504.3 | 3886.4 | 3675.9 | 3500.4 | 3062.3 | 20.0 | **4186.8** |
| Median HNS | 46.7% | 53.3% | 47.7% | 42.9% | 47.3% | 36.8% | -1.5% | **53.6%** |
| Mean HNS | 82.0% | 91.5% | 99.0% | 75.0% | 91.7% | 85.4% | -45.4% | **114.9%** |

# D  CCIL with environment interaction

In this section, we compare CCIL with environment interaction, which employs targeted intervention during environment interaction. Specifically, CCIL infers a causal mask over disentangled latent variables from $\beta$-VAE, by utilizing the returns from environments. As shown in Figure 9, the performance of CCIL improves during environment interaction of 100 episodes, but OREO still exhibits superior performance to CCIL on most confounded Atari environments. This again demonstrates the difficulty of learning disentangled representations from high-dimensional images [28].

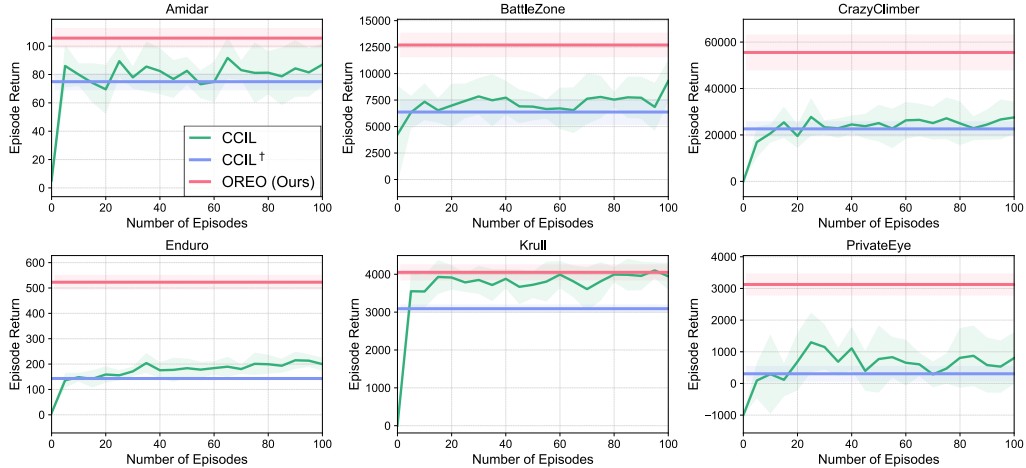

Figure 9: We compare OREO to CCIL with environment interaction, on 6 confounded Atari environments. CCIL[†] denotes the results without environment interaction. The solid line and shaded regions represent the mean and standard deviation, respectively, across eight runs. OREO still outperforms CCIL in most cases, although environment interaction slightly improves the performance of CCIL[†].

# E  Applying OREO to inverse reinforcement learning

We investigate the possibility of applying OREO to other IL methods. While there could be various approaches to utilize the proposed approach for utilizing our regularization scheme for IL, we consider a straightforward application of OREO to a state-of-the-art IL method, i.e., DRIL [8]. Specifically, we apply OREO to the components of DRIL which involves behavioral cloning, i.e., initializing a BC policy and computing rewards with an ensemble of BC policies. In Figure 10, we observe that DRIL + OREO improves the sample-efficiency of DRIL since OREO enables us to learn high-quality BC policies that also result in high-quality reward signals which boosts sample-efficiency. We remark that these results show that IRL methods can also suffer from the causal confusion problem, and a proper regularization scheme can improve the performance by addressing the confusion problem.

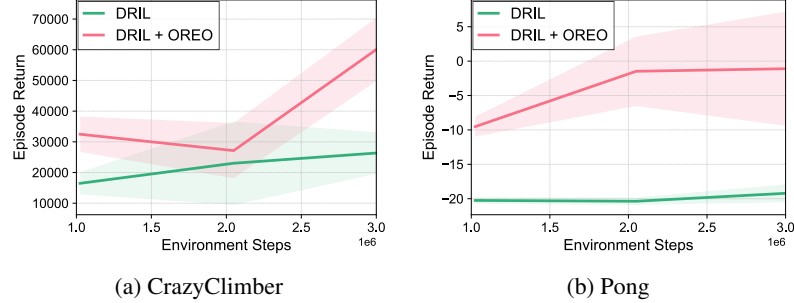

(a) CrazyClimber                  (b) Pong

Figure 10: We apply OREO to the inverse reinforcement learning method (i.e., DRIL [8]) and observe that OREO improves the sample-efficiency of DRIL on confounded CrazyClimber and Pong environments. The solid line and shaded regions represent the mean and standard deviation, respectively, across four runs.

## F  OREO with a sequence of observations

A natural extension of OREO is to apply our regularization scheme to address the causal confusion problem from a sequence of observations [4, 54]. By extracting semantic objects with the same discrete code from consecutive images and dropping the codes from all images, OREO can regularize the policy consistently over multiple images. In this section, we investigate the effectiveness of OREO on such setup by providing additional experimental results on confounded environments where inputs are four stacked observations. Specifically, we mask the features that correspond to the same discrete codes from each observation, and utilize the aggregated masked features for policy learning. In Table 6, we observe that OREO significantly improves the performance of BC, which shows that OREO can also be effective on this setup by regularizing the policy consistently over multiple frames.

Table 6: Performance of policies trained with four stacked observations on 8 confounded Atari environments. The results for each environment report the mean and standard deviation of returns over four runs.

| Environment | BC | Dropout | DropBlock | OREO |
|---|---|---|---|---|
| BankHeist | 448.6$\pm$ 17.8 | 477.6$\pm$ 36.4 | 466.2$\pm$ 17.5 | **538.8**$\pm$ **13.9** |
| Enduro | 167.8$\pm$ 31.7 | 253.1$\pm$ 21.1 | 172.6$\pm$ 18.4 | **426.0**$\pm$ **18.1** |
| KungFuMaster | 13523.5$\pm$ 831.7 | 15041.0$\pm$ 1011.8 | 14859.2$\pm$ 1242.6 | **18375.2**$\pm$ **1055.3** |
| Pong | 4.8$\pm$ 0.9 | 8.2$\pm$ 0.2 | 9.5$\pm$ 0.4 | **12.2**$\pm$ **0.4** |
| PrivateEye | 2349.4$\pm$ 253.1 | 2173.8$\pm$ 168.3 | **2611.4**$\pm$ **476.8** | 2580.7$\pm$ 484.2 |
| RoadRunner | 15189.5$\pm$ 1829.0 | 16574.0$\pm$ 2799.3 | 16901.0$\pm$ 1790.1 | **18726.2**$\pm$ **876.5** |
| Seaquest | 353.4$\pm$ 11.8 | 351.4$\pm$ 28.6 | 315.3$\pm$ 17.7 | **393.2**$\pm$ **19.7** |
| UpNDown | 4075.5$\pm$ 165.6 | 4306.6$\pm$ 216.9 | 4448.9$\pm$ 450.5 | **5193.7**$\pm$ **513.5** |
| Median HNS | 56.6% | 65.1% | 59.3% | **76.7%** |
| Mean HNS | 62.3% | 71.0% | 68.8% | **87.4%** |

## G  Comparison with DropBottleneck

In this section, we compare OREO with DropBottleneck (DB; [22]), which is a dropout-based method that drops features from input variable $X$ redundant for predicting target variable $Y$. While this method was successfully applied to remove the dynamics-irrelevant information such as noises by setting input variable $X$ and target variable $Y$ to two consecutive states, we remark that removing task-irrelevant information cannot be an effective recipe for addressing the causal confusion problem. This is because the causal confusion comes from the difficulty of identifying the true cause of expert actions when both confounders and the causes are strongly correlated with expert actions, i.e., they are both task-relevant information. To support this, we provide experimental results where we jointly optimize DB objective when training a BC policy, i.e., setting the target variable $Y$ to expert actions (denoted as **DB (Y=action)**) in Table 7. In addition, following the original setup in [22], we also provide experimental results where input and target variables are consecutive two states (denoted as **DB (Y=state)**) in Table 8. We observe that **DB (Y=action)** shows comparable performance to OREO in some environments (e.g., CrazyClimber), but OREO still significantly outperforms the suggested baseline in most environments (e.g., Alien, KungFuMaster, and Pong). **DB (Y=state)** performs no better than BC in most environments except for CrazyClimber. These results show that removing dynamics-irrelevant information might not be enough for addressing the causal confusion problem.

Table 7: The results for each environment report the mean and standard deviation of returns over four (DB with expert action) or eight (others) runs. As for the scale of compression term $\beta$ in DB, we choose a better hyperparameter from an array of [0.001, 0.0001].

| Environments | BC | Dropout | DropBlock | DB (Y=action) | OREO |
|---|---|---|---|---|---|
| Alien | 954.1 $\pm$ 83.9 | 1003.8 $\pm$ 53.6 | 926.4 $\pm$ 70.5 | 994.5 $\pm$ 85.6 | **1056.2 $\pm$ 61.6** |
| CrazyClimber | 45372.9 $\pm$ 5508.9 | 39501.6 $\pm$ 6499.3 | 38345.6 $\pm$ 7190.8 | **60996.8 $\pm$ 7943.5** | 55523.4 $\pm$ 7722.2 |
| KungFuMaster | 15074.8 $\pm$ 275.5 | 14452.1 $\pm$ 865.4 | 15753.0 $\pm$ 1265.2 | 15139.5 $\pm$ 867.4 | **18065.6 $\pm$ 1411.5** |
| Pong | 3.2 $\pm$ 0.7 | 10.2 $\pm$ 1.3 | 11.5 $\pm$ 1.3 | 8.2 $\pm$ 0.4 | **14.2 $\pm$ 0.4** |

Table 8: The results for each environment report the mean and standard deviation of returns over four (DB with consecutive state) or eight (others) runs. As for the scale of compression term $\beta$ in DB, we choose a better hyperparameter from an array of [0.001, 0.0001].

| Environments | BC | Dropout | DropBlock | DB (Y=state) | OREO |
|---|---|---|---|---|---|
| Alien | $954.1 \pm 83.9$ | $1003.8 \pm 53.6$ | $926.4 \pm 70.5$ | $896.4 \pm 10.7$ | $\mathbf{1056.2 \pm 61.6}$ |
| CrazyClimber | $45372.9 \pm 5508.9$ | $39501.6 \pm 6499.3$ | $38345.6 \pm 7190.8$ | $\mathbf{60111.5 \pm 5597.8}$ | $55523.4 \pm 7722.2$ |
| KungFuMaster | $15074.8 \pm 275.5$ | $14452.1 \pm 865.4$ | $15753.0 \pm 1265.2$ | $15014.3 \pm 1056.2$ | $\mathbf{18065.6 \pm 1411.5}$ |
| Pong | $3.2 \pm 0.7$ | $10.2 \pm 1.3$ | $11.5 \pm 1.3$ | $3.5 \pm 2.1$ | $\mathbf{14.2 \pm 0.4}$ |

# H A categorical version of CRLR

In this section, we provide a categorical version of Causally Regularized Logistic Regression (CRLR [47]) method. We first formulate the problem setup and briefly introduce some background on CRLR. Given the training data $\mathcal{D} = \left\{ x^{(i)}, y^{(i)} \right\}_{i=1}^{N}$, where $x \in \mathbb{R}^d$ represents the features and $y$ represents labels, the causal classification task targets to jointly identify the causal contribution $\beta \in \mathbb{R}^d$ for all features and learn a classifier $f(\cdot)$ based on $\beta$. As we have no prior knowledge of the causal structure, a reasonable way to adapt causal inference into the classification task is to regard each feature $x_j$ as a treated variable, and all the remaining features $x_{-j} = x \setminus x_j$ as confounding variables, i.e., confounders. To safely estimate the causal contribution of a given feature $x_j$, one has to remove the confounding bias induced by the different distributions of confounders $x_{-j}$ between the treated and control groups. CRLR finds optimal sample weights to balance the distribution of the treated and control group for any treated variable, under an assumption of binary features. To this end, CRLR learns those sample weights by minimizing a causal regularizer as follows:

$$\min_{\left\{ w^{(i)} \right\}} \sum_j \left\| \frac{\sum_{i_0 \in I_{j,0}} w^{(i_0)} x_{-j}^{(i_0)}}{\sum_{i_0 \in I_{j,0}} w^{(i_0)}} - \frac{\sum_{i_1 \in I_{j,1}} w^{(i_1)} x_{-j}^{(i_1)}}{\sum_{i_1 \in I_{j,1}} w^{(i_1)}} \right\|_2^2 ,$$

where $w^{(i)}$ is the sample weight for $x^{(i)}$, and $I_{j,c}$ denotes $\left\{ i \mid x_j^{(i)} = c \right\}$. The original version of CRLR is built upon the binary features, however, it can be naturally extended to a categorical version, by computing the confounder balancing term for any pair of categorical variables. We convert given categorical features $x = [x_1, \cdots, x_d]$ to one-hot encoded binary features $\mathbf{x}$, i.e., $\mathbf{x} = [\mathbf{x}_1, \cdots, \mathbf{x}_d]$ where $\mathbf{x}_i$ is an one-hot encoded version of each feature $x_i$. We denote $\mathbf{x}_{-j} = [\mathbf{x}_1, \cdots, \mathbf{x}_{j-1}, 0, \mathbf{x}_{j+1}, \cdots, \mathbf{x}_d]$ as confounding variables of these one-hot features. Then, a categorical version of the causal regularizer is computed as follows:

$$\min_{\left\{ w^{(i)} \right\}} \sum_j \sum_{\substack{c_1 < c_2 \\ \in \{c \in [K] \mid |I_{j,c}| > 0\}}} \left\| \frac{\sum_{i_1 \in I_{j,c_1}} w^{(i_1)} \mathbf{x}_{-j}^{(i_1)}}{\sum_{i_1 \in I_{j,c_1}} w^{(i_1)}} - \frac{\sum_{i_2 \in I_{j,c_2}} w^{(i_2)} \mathbf{x}_{-j}^{(i_2)}}{\sum_{i_2 \in I_{j,c_2}} w^{(i_2)}} \right\|_2^2 ,$$

where $c_1$, $c_2$ are categorical variables from the set $[K] := \{1, \cdots, K\}$. To apply CRLR on high-dimensional images, we adapt this categorical version on top of VQ-VAE discrete codes. The implementation details of the VQ-VAE are same as OREO (see Appendix B.2). Given a state-action pair $\left( s^{(i)}, a^{(i)} \right)$, a VQ-VAE encoder $g$ represents the state into code indices $q^{(i)} := (q(i, 1), \cdots, q(i, L))$ (see Section 3.1). The one-hot encoded version of the code indices $q$ are denoted as $\mathbf{q}$, similarly to above. Then, a policy $\pi$ and sample weights $\left\{ w^{(i)} \right\}$ are jointly trained by minimizing a weighted behavioral cloning objective and the proposed regularizer:

$$\mathcal{L}_{\text{CRLR}} = \sum_i -w^{(i)} \log \pi \left( a^{(i)} | \mathbf{q}^{(i)} \right)$$

$$+ \lambda \sum_j \sum_{\substack{c_1 < c_2 \\ \in \{c \in [K] \mid |I_{j,c}| > 0\}}} \left\| \frac{\sum_{i_1 \in I_{j,c_1}} w^{(i_1)} \mathbf{q}_{-j}^{(i_1)}}{\sum_{i_1 \in I_{j,c_1}} w^{(i_1)}} - \frac{\sum_{i_2 \in I_{j,c_2}} w^{(i_2)} \mathbf{q}_{-j}^{(i_2)}}{\sum_{i_2 \in I_{j,c_2}} w^{(i_2)}} \right\|_2^2 ,$$

where $\lambda$ is a loss weight for the regularizer. We update $\pi$ and $\left\{ w^{(i)} \right\}$ iteratively until the objective converges, using the gradient descent optimizer.

# I Extended experimental results on confounded Atari environments

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

| Environment | BC | Dropout | DropBlock | Cutout | RandomShift | CCIL† | CRLR | OREO |
|---|---|---|---|---|---|---|---|---|
| Alien | 986.5± 54.4 | 1117.2± 58.8 | 1094.8± 73.7 | 1104.4± 139.5 | 863.5± 68.0 | 1050.4± 62.4 | 100.0± 0.0 | **1222.2**± **95.4** |
| Amidar | 90.8± 7.7 | 81.6± 8.2 | 113.5± 12.9 | 125.0± 7.7 | 78.2± 9.2 | 78.6± 3.1 | 12.0± 0.0 | **130.5**± **16.8** |
| Assault | 816.8± 25.0 | 901.1± 22.6 | 829.9± 23.7 | 694.1± 9.5 | 848.7± 17.0 | 755.5± 9.9 | 0.0± 0.0 | **905.2**± **24.2** |
| Asterix | 249.0± 142.5 | 176.6± 91.4 | 252.2± 139.9 | 195.0± 28.5 | 99.1± 56.6 | 314.1± 7.9 | **592.5**± **148.4** | 212.5± 108.5 |
| BankHeist | 399.0± 22.9 | 476.6± 24.6 | 471.2± 17.8 | 442.5± 20.6 | 354.8± 18.1 | **606.1**± **31.7** | 0.0± 0.0 | 448.4± 13.4 |
| BattleZone | 10933.8± 642.0 | 11621.2± 714.0 | **12067.5**± **1269.0** | 10641.2± 328.5 | 8748.8± 745.8 | 11191.2± 709.5 | 5615.0± 4482.6 | 11703.8± 862.6 |
| Boxing | 21.8± 4.6 | 25.7± 4.1 | 32.1± 5.0 | 21.2± 3.4 | 35.8± 4.3 | 34.2± 2.9 | -43.0± 0.0 | **39.9**± **2.2** |
| Breakout | **6.4**± **0.5** | 2.9± 2.5 | 6.0± 0.9 | 3.1± 2.4 | 4.4± 2.4 | 2.1± 2.0 | 0.0± 0.0 | 5.4± 1.0 |
| ChopperCommand | 1163.0± 129.7 | 1162.0± 51.9 | 1161.8± 64.2 | 1183.9± 56.4 | 1026.2± 83.0 | 1027.2± 78.2 | 1070.2± 10.9 | **1282.9**± **81.1** |
| CrazyClimber | 54142.2± 10143.4 | 54965.4± 6305.6 | 55854.0± 7056.0 | 47456.4± 8129.0 | 60465.9± 9050.9 | 39015.2± 2266.3 | 885.5± 864.8 | **69380.1**± **8907.6** |
| DemonAttack | 238.8± 21.6 | **359.3**± **47.3** | 225.6± 26.1 | 217.8± 20.1 | 294.8± 42.3 | 194.6± 9.3 | 22.7± 41.1 | 0.0± 0.0 |
| Enduro | 226.2± 24.6 | 304.6± 31.4 | 359.1± 38.0 | 132.9± 4.7 | 282.2± 27.4 | 182.8± 6.2 | 0.8± 1.2 | **514.4**± **38.1** |
| Freeway | 32.3± 0.3 | 32.6± 0.2 | 32.6± 0.3 | 32.8± 0.2 | 33.0± 0.3 | **33.1**± **0.2** | 21.4± 0.1 | 32.9± 0.1 |
| Frostbite | 153.6± 20.6 | 149.2± 15.1 | **165.7**± **19.7** | 135.2± 20.1 | 133.2± 33.1 | 96.7± 13.3 | 78.1± 3.4 | 152.7± 23.8 |
| Gopher | 1874.4± 185.8 | 2220.4± 156.2 | 2040.5± 140.2 | 1588.2± 106.1 | 1456.2± 114.2 | 1301.9± 219.5 | 0.0± 0.0 | **2903.9**± **146.6** |
| Hero | 15100.4± 774.6 | 15994.4± 737.5 | 17058.6± 419.4 | 15971.8± 239.4 | 14867.2± 904.5 | **17487.6**± **813.5** | 0.0± 0.0 | 16370.3± 501.4 |
| Jamesbond | 447.6± 33.2 | 492.3± 30.4 | 481.9± 24.6 | 418.9± 15.2 | 452.1± 15.6 | 460.4± 12.5 | 0.0± 0.0 | **527.9**± **20.7** |
| Kangaroo | 3162.8± 209.3 | 2860.4± 175.1 | **3638.6**± **312.6** | 3242.6± 124.2 | 2202.1± 313.5 | 2938.1± 391.6 | 0.0± 0.0 | 3602.9± 189.6 |
| Krull | 4447.9± 91.5 | **4764.7**± **112.3** | 4526.5± 113.7 | 4270.6± 130.6 | 4611.6± 144.9 | 4247.1± 140.0 | 0.0± 0.0 | 4633.6± 114.9 |
| KungFuMaster | 12900.6± 884.3 | 14994.5± 1100.4 | 14819.0± 806.0 | 9956.9± 803.3 | 11698.0± 1330.0 | 12876.9± 912.2 | 0.0± 0.0 | **16955.5**± **1144.2** |
| MsPacman | 1921.9± 174.1 | 2022.6± 202.8 | 2151.7± 178.5 | 1949.7± 176.1 | 1046.3± 220.0 | 1160.6± 144.1 | 70.0± 0.0 | **2263.8**± **165.3** |
| Pong | 3.7± 1.6 | 10.0± 0.8 | 11.6± 0.6 | 7.8± 1.2 | 0.8± 2.1 | -19.8± 0.4 | -21.0± 0.0 | **12.5**± **0.7** |
| PrivateEye | 3035.4± 482.8 | 3396.3± 205.9 | 3057.6± 447.0 | 3092.2± 305.9 | **3578.9**± **222.9** | 1016.4± 286.8 | -1000.0± 0.0 | 3162.6± 282.3 |
| Qbert | 5925.4± 693.9 | **6363.1**± **539.9** | 5904.3± 911.5 | 6174.8± 585.8 | 4100.1± 672.1 | 5056.3± 456.9 | 125.0± 0.0 | 5763.4± 493.4 |
| RoadRunner | 18010.1± 731.1 | 20137.8± 1590.2 | 22522.5± 1749.1 | 12698.9± 1272.2 | 15615.4± 712.1 | 18985.2± 2105.5 | 1528.6± 496.5 | **27303.9**± **2326.7** |
| Seaquest | 527.5± 61.2 | 644.4± 104.2 | 622.3± 79.3 | 376.6± 35.0 | **948.0**± **95.5** | 402.4± 29.3 | 169.8± 54.6 | 921.0± 64.9 |
| UpNDown | 3782.1± 245.7 | 3504.3± 197.1 | 3886.4± 257.1 | 3675.9± 255.0 | 3500.4± 246.8 | 3062.3± 110.3 | 20.0± 0.0 | **4186.8**± **312.0** |
| Median HNS | 46.7% | 53.3% | 47.7% | 42.9% | 47.3% | 36.8% | -1.5% | **53.6%** |
| Mean HNS | 82.0% | 91.5% | 99.0% | 75.0% | 91.7% | 85.4% | -45.4% | **114.9%** |