# OpenReview forum: "Object-Aware Regularization for Addressing Causal Confusion in Imitation Learning"
_NeurIPS.cc/2021/Conference — NeurIPS 2021 Poster_

### Official Review · Reviewer_LhiA · 2021-07-14

**Rating:** 6
**Confidence:** 3

**Summary:**

This paper introduces a policy regularization method that randomly masks objects which share discrete codes inferred from VQ-VAE, to prevent learners from focusing on non-essential parts of expert actions. OREO, the suggested model, is trained without environment interaction but learns how to evenly attend to all objects in confounded Atari games. Consequently, OREO aims to solve the causal confusion problem which is an innate weakness of behavior cloning from observation.

**Limitations And Societal Impact:**

Yes, the authors pointed out agreeable limitations and potential negative societal impacts.
However, it would be interesting to give the difference with CCIL in terms of constructing a causal model(graph), because CCIL proposed the way to deduce the cause-effect relations as a causal graph while OREO did not. I think the dropping method does not lessen the need to know cause-effect relation in causal confusion problems, rather the method can provide a better way to utilize the inferred cause-effect relation.


**Main Review:**

The paper is well written, but it has limited novelty.
1. (originality) OREO can be considered as BC with VQ-VAE, with no other contributions.
It has a better performance compared to CCIL, but the poor performance of CCIL seems due to insufficiency of beta-VAE in a high-dimensional environment, and the better results of OREO are likely to result from the ability of VQ-VAE that infers well discrete latent representations in a high-dimensional environment.
2. (significance) Also, if the authors wanted to claim the suggestion of the dropping technique as a contribution, I think the paper should have included the baseline like DropOut and DropBlock within VQ-VAE architecture, rather than pure convolutional encoder feature maps. To elaborate, the authors suggest the dropping of feature map units that share the same discrete codes, but I think dropping feature map units of random discrete codes or random contiguous discrete codes like in DropOut and DropBlock would be better baselines for the dropping technique.

#### Minor Questions
1. (clarity/ quality) Is there any reason you choose DRIL and GAIL as IRL baselines? GAIL has already been mentioned in CCIL as “GAIL fails to converge to above-chance performance on any of the (confounded) Atari environments.” I wonder what insights that the evaluation result of DRIL and GAIL each can give to the readers.
2. (clarity) Please explain more about Figure 3. For people who don’t have prior knowledge of Atari games, it is hard to know the intention and meaning of even column images.


**Time Spent Reviewing:**

10 hours

---

> ### Author Response · Authors · 2021-08-10
> **Response to Reviewer LhiA**
>
> Dear Reviewer LhiA,
>
> We sincerely appreciate your valuable and insightful comments. We found them extremely helpful for improving our manuscript. We address each comment in detail, one by one below.
>
> ---
>
> **Q1. Novelty of OREO / Reason for the superiority of OREO to CCIL [1]**
>
> **A1.**  As highlighted by Reviewer 8tr4 and Reviewer jmzw, we emphasize that OREO is not the naive combination of BC and VQ-VAE, but a novel method carefully designed to utilize VQ-VAE for addressing the causal confusion problem. In particular, OREO (a) learns a BC policy on top of a trainable convolutional encoder and (b) utilizes the discrete codes from fixed VQ-VAE for regularizing the policy in an object-aware manner (see Figure 2 in the original draft for illustration), instead of learning the policy on top of fixed representations. Hence, unlike CCIL [1] that suffers from the performance bottleneck by utilizing fixed $\beta$-VAE representations as inputs to the policy, OREO can achieve strong performance by avoiding such bottleneck while enjoying the benefit from our novel object-aware regularization.
>
> To address your concern further, we provide additional experimental results that evaluate VQ-VAE + BC, where a policy is learned on top of fixed VQ-VAE representations.
>
> \begin{array}{lcccc}
> \text{Environments} & \text{VQ-VAE + BC} & \text{CCIL}^\dagger & \text{BC} & \text{OREO} \newline
> \hline
> \text{KungFuMaster}       & 11055.1\pm 867.2 & 13394.9\pm 1261.9 & 15074.8\pm 275.5 & \textbf{18065.6}\pm \textbf{1411.5} \newline
> \text{CrazyClimber}         & 33614.9\pm 2183.4 & 22616.8\pm 3282.4 & 45372.9\pm 5508.9 & \textbf{55523.4}\pm \textbf{7722.2} \newline
> \end{array}
> **The results for each environment report the mean and standard deviation of returns over eight runs. CCIL$^\dagger$ denotes the results without environment interaction.**
>
> One can see that the performance of VQ-VAE + BC is worse than vanilla BC, due to the bottleneck from utilizing fixed VQ-VAE representations for learning policies. We also observe that VQ-VAE + BC is worse than CCIL$^\dagger$ that learns a policy on top of fixed $\beta$-VAE representations on confounded KungFuMaster environment, which shows that the superiority of OREO is not just from utilizing VQ-VAE instead of $\beta$-VAE. We will clarify the novelty of OREO in the final draft with relevant experimental results.
>
> ---
>
> **Q2. Comparison with VQ-VAE + Regularization techniques.**
>
> **A2.** Due to a similar reason why VQ-VAE + BC is not that effective (see **Q1** for relevant discussion), we expect that using Dropout and DropBlock within VQ-VAE architecture is also not effective. Nevertheless, following your suggestion, we conducted additional experiments using Dropout and DropBlock within VQ-VAE architecture.
>
> \begin{array}{lccccc}
> \text{Environments} & \text{VQ-VAE + BC} & \text{VQ-VAE + Dropout} & \text{VQ-VAE + DropBlock} & \text{BC} & \text{OREO} \newline
> \hline
> \text{KungFuMaster}       & 11055.1\pm 867.2 & 13323.0\pm 1390.0 & 14861.0\pm 1561.5 & 15074.8\pm 275.5 & \textbf{18065.6}\pm \textbf{1411.5} \newline
> \text{CrazyClimber}         & 33614.9\pm 2183.4 & 21809.9\pm 3781.1 & 18760.8\pm 3470.8 & 45372.9\pm 5508.9 & \textbf{55523.4}\pm \textbf{7722.2} \newline
> \end{array}
> **The results for each environment report the mean and standard deviation of returns over eight runs.**
>
> We observe that OREO still significantly outperforms suggested baselines, which shows that our object-aware regularization technique is more effective than applying Dropout or DropBlock to discrete codes, which is not object-aware.
>
> ---
>
> **Q3. Reason for choosing GAIL and DRIL as baselines?**
>
> **A3.** As noted in our draft, we compare OREO with IRL methods (GAIL and DRIL) in order to show that OREO can exhibit strong performance without environment interaction. To this end, we compare to (a) GAIL, which is widely used as a baseline for various works on IL, and (b) DRIL, which is one of the strongest IRL methods that works on Atari environments. As for the reason why we consider GAIL as our baseline even though [1] reported that GAIL performs worse, we report the performance of GAIL for consistency in the experimental setup, i.e., [1] evaluates GAIL with inputs of 2 stacked frames, but we consider all methods with a single frame as inputs, where GAIL still struggles to achieve strong performance. We will include more explanations in the final draft.
>
> ---
>
> **Q4. More explanation on Figure 3.**
>
> **A4.** The Atari environments in Figure 3 are Frostbite, Pong, Qbert, Gopher, KungFuMaster, BattleZone, Krull, and Boxing (from left to right, top to bottom). As noted in the draft, we observe that similar objects are mapped into the same code. For example, the paddles in Pong environment and the carrots in Gopher environment are assigned to the same code, respectively. These results imply that our method randomly drops similar objects together, therefore regularizes a policy in an object-aware manner. We will incorporate detailed explanations in the final draft for better clarity.
>
> ---
>
> **Q5. More Discussion on the difference with CCIL.**
>
> **A5.** Thanks for the helpful suggestion. As you mentioned, the goal of OREO is not to deduce the cause-effect relation, but to encourage a BC policy to identify the cause of expert actions by regularizing a policy not to exploit nuisance correlates in the visual observations, which is a main difference in the approach of CCIL [1] and OREO. In that sense, we agree that regularizing a policy using OREO is an orthogonal direction to the approach of deducing cause-effect relations, and OREO can be also useful when used with such an approach. We will include more discussion on the difference with CCIL in terms of constructing a causal model in Related work and Discussion sections in the final draft.
>
> ---
>
> **References**
>
> [1] de Haan, Pim, Jayaraman, Dinesh, and Levine, Sergey. Causal confusion in imitation learning. In Advances in Neural Information Processing Systems, 2019.

---

> > ### Comment · Reviewer_LhiA · 2021-08-19
> > **Thanks for the detailed answers to my questions.**
> >
> > Thanks for the detailed answers to my questions. The clarifications convince me well. However, after reviewing the other reviewer's review and the submitted paper, I have one more question.
> >
> >
> > I think the related work [1] is missing in the paper and I suggest the authors consider a comparison with it. As the main contribution of the paper is the combination of VQ-VAE and regularization (i.e. feature drop) technique, comparison with Drop-Bottleneck (DB) [1] seems to be an important baseline since DB also proposes a feature-wise dropout scheme with drop probability $p \sim \text{Bernoulli}(p)$. Furthermore, DB aims for robust exploration (ex. noisy-TV experiment) which is also relevant to the causal confusion problem. The difference between DB and OREO is that DB provides a theoretically sound framework (i.e. valid information bottleneck-based discrete representation learning) and it learns the feature-wise drop probability $p$, while OREO utilizes the VQ-VAE framework for object-aware regularization. Thus, it seems the comparison and discussions with DB should be added.
> >
> > [1] [https://arxiv.org/abs/2103.12300]
> > @misc{kim2021dropbottleneck,
> >       title={Drop-Bottleneck: Learning Discrete Compressed Representation for Noise-Robust Exploration},
> >       author={Jaekyeom Kim and Minjung Kim and Dongyeon Woo and Gunhee Kim},
> >       year={2021},
> >       eprint={2103.12300},
> >       archivePrefix={arXiv},
> >       primaryClass={cs.LG}
> > }

---

> > > ### Author Response · Authors · 2021-08-23
> > > **Additional results for Reviewer LhiA**
> > >
> > > Thanks for your pointer to the DropBottleneck (DB; [1]) paper. While DB is originally designed for robust exploration in noisy environments, we agree with your point that their feature-wise dropout scheme could be considered as our related work.
> > >
> > > Here, we would like to first point out that there is a critical difference between the robust exploration (targeted by [1]) and the causal confusion problem (targeted by our paper). Since [1] considers noises which are task-irrelevant (or dynamics-irrelevant) information that makes it difficult to measure the novelty of states, DB can effectively remove such information by making features from two consecutive states informative to each other and ignoring noises, i.e., setting input variable $X$ and target variable $Y$ in DB to consecutive states. Instead, the causal confusion problem comes from the difficulty of identifying the true cause of expert actions when both confounders and the causes are strongly correlated with expert actions, i.e., they are both task-relevant information.
> > >
> > > To support this, we provide additional experimental results where we jointly optimize DB objective when training a BC policy, i.e., setting the target variable $Y$ to expert actions (denoted as **DB (Y=action)** in the table). We observe that DB (Y=action) shows comparable performance to OREO in some environments (e.g., CrazyClimber), but still OREO significantly outperforms the suggested baseline in most environments (e.g. Alien, Pong, and KungFuMaster), which implies that DB could not effectively remove nuisance correlates from representations in these environments.
> > >
> > > \begin{array}{lccccc}
> > > \text{Environments} & \text{BC} & \text{Dropout} & \text{DropBlock} & \text{DB (Y=action)} & \text{OREO} \newline
> > > \hline
> > > \text{Alien}       &  954.1\pm 83.9 & 1003.8\pm 53.6 & 926.4\pm 70.5 & 994.5\pm 85.6 & \textbf{1056.2}\pm \textbf{61.6} \newline
> > > \text{Pong}         &  3.2\pm 0.7& 10.2\pm 1.3 & 11.5\pm 1.3 & 8.2\pm 0.4 & \textbf{14.2}\pm \textbf{0.4} \newline
> > > \text{KungFuMaster}       & 15074.8\pm 275.5 & 14452.1\pm 865.4 & 15753.0\pm 1265.2 &  15139.5 \pm 867.4 & \textbf{18065.6}\pm \textbf{1411.5} \newline
> > > \text{CrazyClimber}         & 45372.9\pm 5508.9 & 39501.6\pm 6499.3 & 38345.6\pm 7190.8 & \textbf{60996.8}\pm \textbf{7943.5} & 55523.4\pm 7722.2 \newline
> > > \end{array}
> > > **The results for each environment report the mean and standard deviation of returns over four (DB with expert action) or eight (others) runs. As for the scale of compression term $\beta$ in DB, we choose a better hyperparameter from an array of [0.001, 0.0001].**
> > >
> > > One can also consider learning a BC policy by jointly preserving dynamics-relevant information exactly as in [1], because such representations can be implicitly learned to ignore confounders while removing dynamics-irrelevant information. To investigate this approach, we also provide experimental results where we learn a BC policy by jointly optimizing DB objectve with consecutive two states as $X$ and $Y$ (denoted as **DB (Y=state)** in the table). Here, we observe that DB (Y=state) performs no better than BC in most environments except for CrazyClimber, which shows that removing dynamics-irrelevant information might not be enough for addressing the causal confusion problem.
> > >
> > > \begin{array}{lccccc}
> > > \text{Environments} & \text{BC} & \text{Dropout} & \text{DropBlock}& \text{DB (Y=state)} & \text{OREO} \newline
> > > \hline
> > > \text{Alien}       &  954.1\pm 83.9 & 1003.8\pm 53.6 & 926.4\pm 70.5 & 896.4\pm10.7 & \textbf{1056.2}\pm \textbf{61.6} \newline
> > > \text{Pong}         &  3.2\pm 0.7& 10.2\pm 1.3 & 11.5\pm 1.3 & 3.5\pm2.1 & \textbf{14.2}\pm \textbf{0.4} \newline
> > > \text{KungFuMaster}       & 15074.8\pm 275.5 & 14452.1\pm 865.4 & 15753.0\pm 1265.2 &  15014.3\pm 1056.2 & \textbf{18065.6}\pm \textbf{1411.5} \newline
> > > \text{CrazyClimber}         & 45372.9\pm 5508.9 & 39501.6\pm 6499.3 & 38345.6\pm 7190.8 & \textbf{60111.5}\pm \textbf{5597.8} & 55523.4\pm 7722.2 \newline
> > > \end{array}
> > > **The results for each environment report the mean and standard deviation of returns over four (DB with consecutive state) or eight (others) runs. As for the scale of compression term $\beta$ in DB, we choose a better hyperparameter from an array of [0.001, 0.0001].**
> > >
> > > Thanks for your helpful suggestion, and we will include the relevant results on more environments and discussion in the final draft. Please let us know if there are additional comments or questions!
> > >
> > > ---
> > >
> > > **References**
> > >
> > > [1] Kim, Jaekyeom and Kim, Minjung and Woo, Dongyeon and Kim, Gunhee. Drop-Bottleneck: Learning Discrete Compressed Representation for Noise-Robust Exploration. In International Conference on Learning Representations, 2021.

---

> > > > ### Comment · Reviewer_LhiA · 2021-08-24
> > > > **Thanks for the additional experimental results**
> > > >
> > > > Thank you for the additional experimental results.
> > > > I'm satisfied with the answers, and now I think this paper is good enough to be accepted. So, I'd like to raise the score from 5 to 6.

---

> > > > > ### Author Response · Authors · 2021-08-24
> > > > > **Thank you for the response**
> > > > >
> > > > > We are happy to hear that our rebuttal addressed your concerns well.
> > > > >
> > > > > Thank you again for the valuable suggestions and comments, which we believe strengthen our paper.
> > > > >
> > > > > If you have any remaining suggestions or concerns, please let us know!
> > > > >
> > > > > Best, Authors.

---

### Official Review · Reviewer_jmzw · 2021-07-16

**Rating:** 6
**Confidence:** 4

**Summary:**

In this paper, the causal confusion problem in imitation learning (IL) is considered. To tackle the confusion that resulted from not knowing the causal relationship, a two-stage approach is proposed. At the first stage, a VQ-VAE model is trained to generate semantic codes for different objects in the state space. At the second stage, a dropout-based approach is adopted to decouple the semantic confusion among the objects. The contributions of the paper are:
>- Propose a practical approach to solve the causal confusion problem for IL;
>- Conduct extensive experiments under Atari to verify the performance of the proposed approach.

**Limitations And Societal Impact:**

They are adequately addressed.

**Main Review:**

Originally: Good

Addressing the causal confusion problem for IL receives significant attention in recent years. The paper proposes a novel and practical approach to deal with this problem.

Quality: Overall good

The paper proposes an interesting approach, which is quite practical in my view. The experiments show significant improvement over existing baselines. While I still think the following limitations exist:
>- The proposed approach seems to work only in vision-based tasks.
>- The effectiveness of the proposed approach seems to rely heavily on the performance of the VQ-VAE model. I am curious to see how the proposed approach is robust to the quality of the semantic codes learned by VQ-VAE.
>- The experiments include only Atari tasks.
>- Only behavior cloning is discussed. While I think applying the approach to other IL methods is also possible. I am curious to see whether this is true.

Clarity: The paper is overall well-written. Both the technical and the experimental parts are clear to be understood.

Significance: To my understanding, the paper provides an interesting approach to address the causal confusion problem in IL, which may be of interest to researchers in this area. I suggest releasing the code to make the results easier to reproduce by other researchers.




**Time Spent Reviewing:**

2 hours

---

> ### Author Response · Authors · 2021-08-10
> **Response to Reviewer jmzw**
>
> Dear Reviewer jmzw,
>
> We sincerely appreciate your valuable and insightful comments. We found them extremely helpful for improving our manuscript. We address each comment in detail, one by one below.
>
> ---
>
> **Q1. Applicability of OREO to state-based environments.**
>
> **A1.** As you pointed out, OREO is not applicable to state-based environments as OREO is designed to regularize a policy in an object-aware manner from high-dimensional observations like images. However, as you also pointed out, we still believe that OREO can be a practical solution to the causal confusion problem in various image-based applications, e.g., video games, self-driving, and robotic manipulation. We will include relevant discussion in the Discussion section of the final draft.
>
> ---
>
> **Q2. Dependence on VQ-VAE.**
>
> **A2.** While the performance of OREO might degrade in extreme cases, e.g., if code assignment of VQ-VAE collapses to one code, we would like to emphasize that we can feasibly obtain high-quality discrete codes even from high-resolution, complex real-world images, as demonstrated in recent works that learn discrete codes, e.g., VQ-VAE2 [1], VQ-GAN [2]. Also, there have been several works that are built upon those discrete codes of VQ-VAE for downstream tasks, e.g., Video-GPT [3], DALL-E [4], which show state-of-the-art performance. Hence, we expect that OREO can be applied to various applications, e.g., self-driving (see **Q3** for supporting experiments), without being bottlenecked by the quality of VQ-VAE.
>
> ---
>
> **Q3. Tasks other than Atari.**
>
> **A3.** To address your concern, we provide additional experimental results on the self-driving CARLA environment [5] (suggested by reviewer 8tr4), where visual observations are more complex and high-resolution than Atari environments we considered. Specifically, we apply OREO to conditional imitation learning in [6] using 150 expert demonstrations from the dataset [7], and compare with Dropout and DropBlock as baselines. We observe that OREO improves the performance of BC, which shows that regularizing a policy in an object-aware manner could be also effective on more complex real-world applications.
>
> \begin{array}{lcccc}
> \text{Task} & \text{BC} & \text{Dropout} & \text{DropBlock} & \text{OREO} \newline
> \hline
> \text{Straight}       & 75.0\pm1.73  & 82.0\pm8.25 & 74.0\pm3.46  & \textbf{87.0} \pm \textbf{4.36} \newline
> \text{One Turn}     & 43.0\pm9.11 & 59.0\pm3.32 & 53.0\pm5.20  & \textbf{70.0} \pm \textbf{7.21} \newline
> \end{array}
> **Performance of policies trained on 150 expert demonstrations from the CARLA driving dataset, under a weather condition of *daytime*. The results for each environment report the mean and standard deviation of success rates over four runs.**
>
> ---
>
> **Q4. Application of OREO to other IL methods.**
>
> **A4.** Thanks for the suggestion to investigate the applicability of OREO to other IL methods. We first remark that we mainly discuss BC since IRL methods might not suffer from the causal confusion problem as they keep collecting new samples from environment interaction (we refer to [8] for relevant discussion). However, OREO can still be applicable to various methods that utilize behavioral cloning [9,10,11].
>
> To support this, we report the results of DRIL + OREO on the confounded Atari environment, where OREO is applied to the components of DRIL which involves behavioral cloning, i.e., initializing a policy with BC policy and computing rewards with an ensemble of BC policies.
>
> **Confounded Atari Pong:**
> \begin{array}{lccc}
> \text{Method}  & & \text{Environment Steps} {(\times10^6)} & \newline
> &  1 & 2 & 3 \newline
> \hline
> \text{DRIL}      & -20.250\pm0.54 & -20.369\pm0.62 & -19.125\pm1.56 \newline
> \text{DRIL + OREO}      & \textbf{-9.525}\pm\textbf{1.448} & \textbf{-1.475}\pm\textbf{5.049} & \textbf{-2.325}\pm\textbf{10.43} \newline
> \end{array}
> **The results report the mean and standard deviation over four runs.**
>
> **Confounded Atari CrazyClimber:**
> \begin{array}{lccc}
> \text{Method}  & & \text{Environment Steps} {(\times10^6)} & \newline
> &  1 & 2 & 3 \newline
> \hline
> \text{DRIL}      & 16487.5\pm4181.3 & 23025.0\pm15696.3 & 26610.0\pm7211.7  \newline
> \text{DRIL + OREO}      & \textbf{31715.0}\pm\textbf{6466.4} & \textbf{31468.8}\pm\textbf{15827.6} & \textbf{53648.9}\pm\textbf{26492.9} \newline
> \end{array}
> **The results report the mean and standard deviation over four runs.**
>
> We observe that DRIL + OREO significantly improves the sample efficiency of DRIL since OREO enables us to learn high-quality BC policies that also results in high-quality reward signal which boosts sample-efficiency. We will include relevant discussion and more experimental results in the final draft.
>
> ---
>
> **Q5. Public source codes.**
>
> **A5.** Thanks for the suggestion. We promise to release the code with the final draft, with improved visibility for easier reproduction of our experimental results.
>
> ---
>
> **References**
>
> [1] Razavi, Ali, Aaron van den Oord, and Oriol Vinyals. "Generating diverse high-fidelity images with vq-vae-2." In Advances in neural information processing systems, 2019
>
> [2] Esser, Patrick, Robin Rombach, and Bjorn Ommer. "Taming transformers for high-resolution image synthesis." In Proceedings of the IEEE/CVF Conference on Computer Vision and Pattern Recognition, 2021
>
> [3] Yan, Wilson, Yunzhi Zhang, Pieter Abbeel, and Aravind Srinivas. "VideoGPT: Video Generation using VQ-VAE and Transformers." arXiv preprint, 2021
>
> [4] Ramesh, Aditya, Mikhail Pavlov, Gabriel Goh, Scott Gray, Chelsea Voss, Alec Radford, Mark Chen, and Ilya Sutskever. "Zero-shot text-to-image generation." arXiv preprint, 2021
>
> [5] Dosovitskiy, Alexey, German Ros, Felipe Codevilla, Antonio Lopez, and Vladlen Koltun. "CARLA: An open urban driving simulator." In Conference on robot learning, 2017
>
> [6] Codevilla, Felipe, Matthias Müller, Antonio López, Vladlen Koltun, and Alexey Dosovitskiy. "End-to-end driving via conditional imitation learning." In 2018 IEEE International Conference on Robotics and Automation (ICRA), 2018.
>
> [7] https://github.com/carla-simulator/imitation-learning
>
> [8] de Haan, Pim, Jayaraman, Dinesh, and Levine, Sergey. Causal confusion in imitation learning. In Advances in Neural Information Processing Systems, 2019.
>
> [9] Ho, Jonathan and Ermon, Stefano. Generative adversarial imitation learning. In Advances in Neural Information Processing Systems, 2016.
>
> [10] Brantley, Kiante, Sun, Wen, and Henaff, Mikael. Disagreement-regularized imitation learning. In International Conference on Learning Representations, 2020.
>
> [11] Rajeswaran, Aravind, Vikash Kumar, Abhishek Gupta, Giulia Vezzani, John Schulman, Emanuel Todorov, and Sergey Levine. "Learning complex dexterous manipulation with deep reinforcement learning and demonstrations." In Robotics: Science and Systems, 2018.

---

> > ### Comment · Reviewer_jmzw · 2021-08-20
> > **discussion on Q4**
> >
> > Thanks for the detailed response. The new experimental results are interesting, and I suggest putting them in the revision.
> >
> > For the following discussion in A4:
> > "we mainly discuss BC since IRL methods might not suffer from the causal confusion problem as they keep collecting new samples from environment interaction"
> >
> > I do not agree with this. In IRL, even though the agent generates new samples in the environment, it uses its own policy. Thus the generated samples still have distributional shifts with the expert demonstrations. I think the causal confusion problem could only be solved when the agent knows what is correct and what is wrong in the shifted distribution, by obtaining additional information such as new demonstrations, environmental rewards, or counterfactual reasoning.

---

> > > ### Author Response · Authors · 2021-08-23
> > > **Discussion on Q4**
> > >
> > > Thanks for your insightful comment. Regarding the reason why we stated 'IRL methods might not suffer from the causal confusion problem' in our response, we again would like to refer to [8] where the authors show that GAIL does not suffer from the causal confusion problem in some setup (e.g., Hopper in OpenAI Gym) when a substantial amount of environment interactions are available.
> > >
> > > However, we agree with your point that such results do not hold in general, i.e., the causal confusion problem cannot be solved completely by just collecting more samples, as there is no additional information for identifying the cause of expert actions and reward function is just learned using the given expert demonstrations where nuisance correlates exist. And we expect this might be the reason why applying OREO to the BC components of DRIL can improve the performance in our experiments on confounded Atari environments.
> > >
> > > P.S. Here, one may suggest to apply regularization techniques (i.e., Dropout, DropBlock, and OREO) to the RL components of DRIL, e.g., learning actor/critic. We actually tried this, but observe that all techniques make training highly unstable, which implies that more careful design for regularization is required. It would be an interesting future direction to develop a careful design that extends our object-aware regularization for addressing the causal confusion in IRL.
> > >
> > > We will include relevant discussions and results in the final draft. Please let us know if you have any more questions or comments!

---

### Official Review · Reviewer_qCdW · 2021-07-16

**Rating:** 6
**Confidence:** 4

**Summary:**

The paper considers a recently identified problem in imitation learning, that the trained policy learns relationships to nuisance variables, which do not have a casual relationship to the main goal of the action.

For instance, when learning pong, the learned policy becomes dependent on the score display and the learned policy is difficult to transfer to an environment where the score display does not exist. On the other hand, a policy learned on an environment where the score display was masked out transfers better to an environment where it is present.

The authors are proposing a technique to regularize the imitation learning process by masking out certain objects from the environment. They achieve this by training a VQ-VAE, and assuming that units of a feature map corresponding to the same objects are mapped to similar discrete codes. The authors then randomly drop the units of a feature map that share the same discrete code during training.

The authors test the proposed approach on a number of games from the Atari environment using the "confounded input model", where the action from the previous time step is encoded into the current input. Previous work had shown that such a confounded input, although it contains more information than the original form of the input, is in practice harder to learn.

**Ethical Concerns:**

No ethical concerns had been found with this paper.


**Limitations And Societal Impact:**

The paper correctly identifies that the proposed work only focuses on the nuisance features on the current state, and does not actually infer the causal relationships.

The authors also discuss some hypothetical negative impacts about self-driving cars and imitation learning, however these are hypothetical and not specific to the contributions of this paper.

**Main Review:**

The paper describes a regularization approach from the general class of approaches initiated by dropout, for the specific problem of nuisance variables in imitation learning. The approach is based on randomly dropping certain objects from the environment, with the objects being identified by the fact that they map to similar discrete codes when encoded by a VQ-VAE.

The paper shows improvement in the performance of the Atari games considered.

There is a certain degree of concern about the significance of the proposed approach. The approach does not strictly speaking identify causal relationships. It seems that the primary impact is that it prevents the policy to focus too much on one particular variable. It is also not necessarily clear why training on the confounded input changes the learned policy, and why the proposed regularization improves it (eg Figure 8). While I am aware that this technique of confusing the input was not proposed in this paper, its practical significance is unclear. Would this problem appear in a real world setting? Would the proposed approach work with real world images?

**Time Spent Reviewing:**

2 hrs

---

> ### Author Response · Authors · 2021-08-10
> **Response to Reviewer qCdW**
>
> Dear Reviewer qCdW,
>
> We sincerely appreciate your valuable and insightful comments. We found them extremely helpful for improving our manuscript. We address each comment in detail, one by one below.
>
> ---
>
> **Q1. OREO on real-world application.**
>
> **A1.** We first remark that the causal confusion problem appears in a real-world setting. For example, [1] showed that a self-driving agent could misidentify a brake indicator as the cause of expert action, instead of attending to a true cause like a pedestrian in front of a car (see Figure 1 of [1] for relevant discussion).
>
> In order to show that OREO is also effective on such real-world applications, we evaluate OREO on the self-driving CARLA environment [2], where visual observations are more complex and high-resolution than Atari environments we considered. Specifically, we apply OREO to conditional imitation learning in [3] using 150 expert demonstrations from the dataset [4], and compare with Dropout and DropBlock as baselines. We observe that OREO improves the performance of BC, which shows that regularizing a policy in an object-aware manner could be also effective on more complex real-world applications.
>
> \begin{array}{lcccc}
> \text{Task} & \text{BC} & \text{Dropout} & \text{DropBlock} & \text{OREO} \newline
> \hline
> \text{Straight}       & 75.0\pm1.73  & 82.0\pm8.25 & 74.0\pm3.46  & \textbf{87.0} \pm \textbf{4.36} \newline
> \text{One Turn}     & 43.0\pm9.11 & 59.0\pm3.32 & 53.0\pm5.20  & \textbf{70.0} \pm \textbf{7.21} \newline
> \end{array}
> **Performance of policies trained on 150 expert demonstrations from the CARLA driving dataset, under a weather condition of *daytime*. The results for each environment report the mean and standard deviation of success rates over four runs.**
>
>
> ---
>
> **Q2. Significance of OREO.**
>
> **A2.** While OREO does not aim to directly identify a causal relationship as you mentioned, we believe OREO is a practical and significant method to address the causal confusion problem, as highlighted by Reviewer jmzw. In particular, we emphasize that OREO encourages a policy to identify the cause of expert actions by regularizing a policy not to exploit nuisance correlates for predicting actions, which is demonstrated to be effective by our extensive experimental results on confounded Atari environments and newly updated self-driving CARLA environment.
>
> ---
>
> **Q3. More discussion on negative societal impacts.**
>
> **A3.** Thanks for the suggestion. We will include more discussion on the societal impacts which are more relevant to our contribution, e.g., environmental impact that could be caused by pre-training of large VQ-VAE models. We will also make it more clear how societal impacts we discussed are relevant to our proposed method and problem setup in the final draft, including the discussion related to the newly updated self-driving experiments on CARLA environment.
>
> ---
>
> **References**
>
> [1] de Haan, Pim, Jayaraman, Dinesh, and Levine, Sergey. Causal confusion in imitation learning. In Advances in Neural Information Processing Systems, 2019.
>
> [2] Dosovitskiy, Alexey, German Ros, Felipe Codevilla, Antonio Lopez, and Vladlen Koltun. "CARLA: An open urban driving simulator." In Conference on robot learning, 2017
>
> [3] Codevilla, Felipe, Matthias Müller, Antonio López, Vladlen Koltun, and Alexey Dosovitskiy. "End-to-end driving via conditional imitation learning." In 2018 IEEE International Conference on Robotics and Automation (ICRA), pp. 4693-4700. IEEE, 2018.
>
> [4] https://github.com/carla-simulator/imitation-learning

---

### Official Review · Reviewer_8tr4 · 2021-07-21

**Rating:** 6
**Confidence:** 3

**Summary:**

In this paper, the authors propose a novel imitation learning method, regularized by object-aware encoding and masking during the training phase. The improvements over the performance of behavioral cloning baselines on Atari games indicate the effectiveness of the proposed framework.

**Ethics Review Area:**

["I don’t know"]

**Limitations And Societal Impact:**

The proposed OREO method heavily utilizes VQ-VAE to disentangle the raw RGB input into discrete codes for each semantic object. So the accuracy of such a grouping could be crucial. Even though VQ-VAE works well on Atari games, the real-world application, such as autonomous driving simulation CARLA, typically has much more complicated scenes with higher resolution. I wonder if the proposed OREO could still help when VQ-VAE generated discrete codebook becomes fuzzier. Also, as mentioned in section 5, the application of learning from a sequence of observations could be interesting to study.

**Main Review:**

- The proposed method is easy to follow. The overall writing looks well-organized to me. Both big picture and implementation details are discussed intensively. I believe the readers could catch up with the most critical observations in an easy way.
- The illustrations are helpful for the readers to realize the main motivations and contributions of the proposed method. One possible suggestion is to highlight the masked area in figure 1. It takes me a long time to realize where the masked area is.

**Time Spent Reviewing:**

3

---

> ### Author Response · Authors · 2021-08-10
> **Response to Reviewer 8tr4**
>
> Dear Reviewer 8tr4,
>
> We sincerely appreciate your valuable and insightful comments. We found them extremely helpful for improving our manuscript. We address each comment in detail, one by one below.
>
> ---
>
> **Q1. OREO on real-world application.**
>
> **A1.** Following your suggestion, we provide additional experimental results on CARLA environment [1] where visual observations are more complex and high-resolution than Atari environments we considered. Specifically, we apply OREO to conditional imitation learning in [2] using 150 expert demonstrations from the dataset [3], and compare with Dropout and DropBlock as baselines. We observe that OREO improves the performance of BC, which shows that regularizing a policy in an object-aware manner could be also effective on more complex real-world applications.
>
> \begin{array}{lcccc}
> \text{Task} & \text{BC} & \text{Dropout} & \text{DropBlock} & \text{OREO} \newline
> \hline
> \text{Straight}       & 75.0\pm1.73  & 82.0\pm8.25 & 74.0\pm3.46  & \textbf{87.0} \pm \textbf{4.36} \newline
> \text{One Turn}     & 43.0\pm9.11 & 59.0\pm3.32 & 53.0\pm5.20  & \textbf{70.0} \pm \textbf{7.21} \newline
> \end{array}
> **Performance of policies trained on 150 expert demonstrations from the CARLA driving dataset, under a weather condition of *daytime*. The results for each environment report the mean and standard deviation of success rates over four runs.**
>
> We would also like to remark that we can obtain high-quality discrete codes even from more high-resolution, complex real-world images, as demonstrated in recent works that learn discrete codes, e.g., VQ-VAE-2 [4], VQ-GAN [5]. Hence, we expect OREO can be also applied to more various applications.
>
> ---
>
> **Q2. OREO with a sequence of observations.**
>
> **A2.** We agree that applying OREO to the setup where inputs are a sequence of observations is an interesting direction, and OREO is naturally applicable. For example, one can consider masking the features of each observation from a convolutional encoder, and utilizing the aggregated masked features for policy learning. While we could not run the experiments due to limited resources an the time constraint, we will continue our effort to provide the results during the discussion phase or in the final draft.
>
> ---
>
> **Q3. Visibility of masks in Figure 1.**
>
> **A3.**  Thanks for pointing this out. We will make masks more visible in the final draft.
>
> ---
>
> **References**
>
> [1] Dosovitskiy, Alexey, German Ros, Felipe Codevilla, Antonio Lopez, and Vladlen Koltun. "CARLA: An open urban driving simulator." In Conference on robot learning, 2017
>
> [2] Codevilla, Felipe, Matthias Müller, Antonio López, Vladlen Koltun, and Alexey Dosovitskiy. "End-to-end driving via conditional imitation learning." In 2018 IEEE International Conference on Robotics and Automation (ICRA), pp. 4693-4700. IEEE, 2018.
>
> [3] https://github.com/carla-simulator/imitation-learning
>
> [4] Razavi, Ali, Aaron van den Oord, and Oriol Vinyals. "Generating diverse high-fidelity images with vq-vae-2." In Advances in neural information processing systems, 2019
>
> [5] Esser, Patrick, Robin Rombach, and Bjorn Ommer. "Taming transformers for high-resolution image synthesis." In Proceedings of the IEEE/CVF Conference on Computer Vision and Pattern Recognition, 2021

---

> ### Author Response · Authors · 2021-08-11
> **Additional results for Reviewer 8tr4**
>
> Dear Reviewer 8tr4,
>
> In addition to our initial response, we now provide supporting results for **A2**; thank you for your patience!
>
> We conducted additional experiments using stacked observations as inputs on the confounded Atari environment. We observe that OREO significantly improves the performance of BC, which shows that OREO can be also effective on this setup by regularizing the policy consistently over multiple timesteps. We will include more experimental results in the final draft.
>
> \begin{array}{lcccc}
> \text{Environment} & \text{BC} & \text{Dropout} & \text{DropBlock} & \text{OREO} \newline
> \hline
> \text{KungFuMaster}     & 13523.5
> \pm831.7
>  & 15041.0
> \pm1011.8
>  & 14859.3
> \pm1242.6
>   & \textbf{18375.3
> } \pm \textbf{1055.3
> } \newline
> \text{CrazyClimber}       & 63905.8
> \pm3679.1
>   & 64799.3
> \pm3770.9
>  & 72056.3
> \pm4025.9
>   & \textbf{72615.0
> } \pm \textbf{2918.8
> } \newline
> \end{array}
> **Performance of policies trained with four stacked observations on confounded Atari environments. The results for each environment report the mean and standard deviation of returns over four runs.**
>
> Sincerely,
> Authors.

---

### Author Response · Authors · 2021-08-18
**A gentle reminder**

Dear Reviewers,

Thank you for your time and efforts in reviewing our paper.

We kindly remind that we are more than one week into the discussion period. We believe that we sincerely and successfully address your concerns/questions/misunderstandings/suggestions, with the results of the supporting experiments.

If you have any further concerns or questions, please do not hesitate to let us know.

Thank you very much! Authors

---

### Decision · Program_Chairs · 2021-09-27

**Decision:**

Accept (Poster)

**Comment:**

This submission targets the problem of "causal confusion" in imitation learning, where spurious correlates in expert data hurt environmental performance. The approach proposed is simple: train a vector-quantized VAE representation, and apply dropout over it during imitation learning. The motivation is: VQ-VAEs can learn to disentangle objects in images, and this can help coherently omit semantic information.

The key strengths of this submission are the simplicity of the proposed approach, the breadth of experimental evaluation, and the strong empirical results. In particular, the evaluation uses 27 Atari environments, aside from new rebuttal experiments in simple CARLA settings. These may become valuable contributions to studying this problem.

This submission is very much borderline, and I only lightly lean towards acceptance. My main reasons for hesitation are:
- Like reviewers have also commented, the paper relies on learning not just good quality discrete codes, but disentangled *objects* through VQ-VAEs (or at least, it is motivated in this way). While this works well enough on Atari settings such as in the submission, I'm not convinced that it will in more complex settings.
- The CARLA settings tested in new rebuttal experiments ("straight", and "one turn") are simple, and not the standard town driving environments commonly used in the imitation learning literature.
- Dropout has previously been proposed as an approach to learn robust imitation policies that can tackle these problems: see [1]. This means that the proposed approach is effectively a combination of VQ-VAE with an existing technique
- Aside from new somewhat preliminary rebuttal experiments on 2 Atari environments in the "stacked inputs" setting, all experiments are performed in the more contrived setting in which past actions are overlaid on the input images. Other works in this literature have now moved away from this assumption, such as [1, 2].

[1] "Chauffeurnet: Learning to drive by imitating the best and synthesizing the worst."
[2] "Fighting Copycat Agents in Behavioral Cloning from Observation Histories"